# Mechanism of the electroneutral sodium/proton antiporter PaNhaP from transition-path shooting

Kei-ichi Okazaki [1,2], David Wöhlert[3], Judith Warnau[2], Hendrik Jung[2], Özkan Yildiz [3], Werner Kühlbrandt[3] & Gerhard Hummer [2,4]

Na$^+$/H$^+$ antiporters exchange sodium ions and protons on opposite sides of lipid membranes. The electroneutral Na$^+$/H$^+$ antiporter NhaP from archaea *Pyrococcus abyssi* (PaNhaP) is a functional homolog of the human Na$^+$/H$^+$ exchanger NHE1, which is an important drug target. Here we resolve the Na$^+$ and H$^+$ transport cycle of PaNhaP by transition-path sampling. The resulting molecular dynamics trajectories of repeated ion transport events proceed without bias force, and overcome the enormous time-scale gap between seconds-scale ion exchange and microseconds simulations. The simulations reveal a hydrophobic gate to the extracellular side that opens and closes in response to the transporter domain motion. Weakening the gate by mutagenesis makes the transporter faster, suggesting that the gate balances competing demands of fidelity and efficiency. Transition-path sampling and a committor-based reaction coordinate optimization identify the essential motions and interactions that realize conformational alternation between the two access states in transporter function.

[1] Department of Theoretical and Computational Molecular Science, Institute for Molecular Science, National Institutes of Natural Sciences, Okazaki 444-8585, Japan. [2] Department of Theoretical Biophysics, Max Planck Institute of Biophysics, 60438 Frankfurt am Main, Germany. [3] Department of Structural Biology, Max Planck Institute of Biophysics, 60438 Frankfurt am Main, Germany. [4] Institute of Biophysics, Goethe University Frankfurt, 60438 Frankfurt am Main, Germany. Correspondence and requests for materials should be addressed to K.-i.O. (email: keokazaki@ims.ac.jp) or to G.H. (email: gerhard.hummer@biophys.mpg.de)

Cation-proton antiporters (CPA) are secondary-active transporters involved in a wide range of cellular processes, from controlling pH and salt concentration to osmoregulation of the cell volume[1,2]. Recently, structures of the archaeal Na$^+$/H$^+$ antiporters PaNhaP from *Pyrococcus abyssi* and MjNhaP1 from *Methanocaldococcus jannaschii* have been solved[3,4]. The archaeal PaNhaP and MjNhaP1 as well as human NHE1, which is linked to a wide spectrum of diseases from heart failure to autism[5] and has no structure solved yet, are electro-neutral antiporters of the CPA1 family, exchanging one proton against one sodium ion. By contrast, the bacterial antiporters NhaA[2,6] and NapA[7] are electrogenic members of the CPA2 family of antiporters, exchanging two protons against one sodium ion. In this paper, we study the transport mechanism of the electroneutral antiporter PaNhaP. Ben-Tal and colleagues recently reported a broad phylogenetic analysis of cation/proton antiporter families CPA1 and CPA2, and identified a motif that determines electrogenicity and ion selectivity[8]. The shared motif determining electrogenicity and ion selectivity[8] supports the use of PaNhaP as a model system in mechanistic studies of electro-neutral Na$^+$/H$^+$ exchange[8,9].

Transporters generally work by an alternating-access mechanism[10]. Switching between inward-open and outward-open conformations alternatingly exposes the binding site to opposite sides of the membrane. To elucidate the transport mechanism in atomic detail, we performed extensive (~ 19 µs) molecular dynamics (MD) simulations. By adapting the transition-path sampling algorithm[11] to shooting trajectories from pre-defined transition regions[12], we determined continuous and unbiased transition trajectories between access states. These transition paths are the exceptional segments of equilibrium trajectories when rare events—here the exchange of ions—actually occur, and contain the mechanistic information about complex molecular processes[11–14]. During the orders-of-magnitude longer times between transition paths, the system simply waits for a rare fluctuation that triggers the escape from its metastable state. With transition-path shooting, we can overcome the huge gap between the microsecond time-scale of MD and the seconds time-scale of ion exchange measured for PaNhaP at room temperature[3]. By resolving transition paths and associated transition-state conformations, we gain direct mechanistic insight into the substrate translocation process. In effect, path sampling allows us to "watch" the transport cycle unfold, with H$^+$ and Na$^+$ as substrates, both at near-ambient temperature (37 °C) and at 100 °C, the physiological temperature for *P. abyssi*.

## Results

### Inward- and outward-open conformations of PaNhaP.
The crystal structures of PaNhaP and MjNhaP1 show an inward-open conformation[3,4], with cations resolved in the ion-binding site of PaNhaP[3]. PaNhaP is thus the only CPA antiporter with a well-characterized ion-binding site[3] (Fig. 1a), making it an ideal candidate for mechanistic characterization. An outward-open structure was obtained by electron cryo-crystallography at low resolution for MjNhaP1[4]. However, using this structure (PDB ID: 4D0A) as reference for targeted MD (TMD) simulations to create an outward-open PaNhaP led to an incomplete conformational change. Thus, to initiate path sampling of the functional anti-porter cycle, we fitted the inward-open crystal structure of MjNhaP1 by MDFF flexible fitting[15] into the 3D EM map of the outward-facing MjNhaP1[4]. We then used the refitted outward-open MjNhaP1 model as a reference to construct the outward-open conformation of PaNhaP by TMD simulations[16]. Similar approaches of using homologous structures as reference for TMD have been applied to other transporters[17,18]. Microsecond-long

unrestrained MD simulations confirmed the stability of the outward-open conformation (Supplementary Figs. 1 and 2). Figure 1b shows the average structure of PaNhaP during this equilibrium trajectory together with the average water density. The ion-binding site in chain A is accessible from the inside, while in chain B it is accessible from the outside (see MD simulations in Methods). By using a symmetrized target structure in TMD simulations, we also obtained a symmetric outward-open PaNhaP dimer conformation (Supplementary Fig. 3). The motions of its two transporter domains closely mirror those of protomer B in the asymmetric structure (Supplementary Figs. 3–5). A translation of the transporter domain normal to the membrane ($\Delta z \sim 3.5$ Å), and a rotation around an axis in the membrane plane ($\Delta \phi \sim 10°$) distinguish the relaxed inward- and outward-open conformations (Fig. 1c, d), suggesting a rigid-body motion that combines elements of the elevator and rocker-switch mechanisms[7,19,20] in a "rocking elevator" transition. The asymmetric dimer conformation obtained in this way is available in PDB format (Supplementary Data 1).

### Transition-path shooting simulations.
The transition dynamics between the inward- and outward-open states is central to the transport mechanism, revealing at once characteristic domain motions, rate-limiting steps, substrate pathways, and the opening and closing of the gate preventing ion leakage. However, with ion exchange occurring on a timescale of seconds at ambient conditions[3,4], regular MD simulations are far too slow to resolve transitions. Furthermore, methods such as TMD give us only a rough idea of the relevant functional motions, since they create pathways under a strong external bias[16]. To sample unbiased transition paths between the inward- and outward-open states, we used an iterative transition-path shooting algorithm[12]. Starting from candidate transition states, conjugate trajectory pairs are shot off "forward" and "backward" in time, simply by sign-inverting initial velocities in one of the segments[11,12]. Pairs of trajectories reaching opposite access states are then seamlessly stitched together into a continuous transition trajectory by inverting the sequence of time in one segment. These transition paths are proper representatives of the ensemble of transition paths that would be obtained in an infinitely long simulation. By microscopic time reversibility, transition paths are identical for the forward (in-out) and reverse (out-in) processes, after reverting time, even in the presence of gradients across the membrane[21].

We performed two independent transition-path-sampling simulations initiated from different initial paths to test for possible bias. As starting paths we used two TMD trajectories, one from the inward-open to the outward-open conformation of PaNhaP, as described above ("first initial path"), and the other a TMD trajectory in the reverse direction, initiated from the outward-open conformation after 1µs equilibration ("second initial path") to test for possible bias. We initiated path sampling from structures along these TMD trajectories. In four rounds of trajectory shooting, we found 22 distinct transition paths (Supplementary Table 1).

Figure 2 shows representative transition paths for H$^+$ and Na$^+$ transport at 37 °C and 100 °C projected onto the plane spanned by the two structural order parameters, $\Delta \phi = (\Delta \phi_{in} - \Delta \phi_{out}) / 2$ and $\Delta z = (\Delta z_{in} + \Delta z_{out}) / 2$. These projections resolve the dominant global motions of the transporter domain in terms of its rigid-body rotation and translation relative to inward- and outward-open structures (see Supplementary Figs. 6–9 for all transition paths obtained). We also determined a hydration order parameter $n_{access}$ that indicates whether the ion-binding site is connected by water to the outside ($n_{access} > 0$) or to the inside

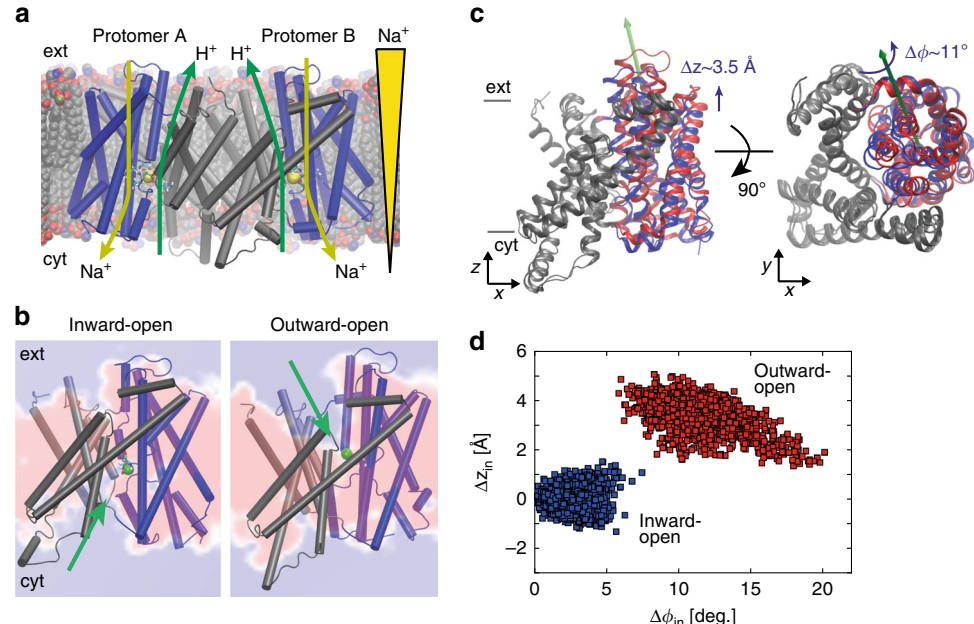

**Fig. 1** Alternating-access transport in PaNhaP. **a** Inward-open dimer with bound Na+ ions (yellow spheres) embedded in a lipid bilayer. Arrows indicate gradient-driven Na+ import (yellow) coupled to H+ export (green) from the cytosol (bottom) to the exterior (top). **b** Inward- and outward-open structures of protomers A and B, respectively, with a section through the average water density in the 1-μs free MD simulation of the asymmetric dimer (red: no water; blue: water at bulk density). The green sphere indicates Hδ of the protonated Asp159, and the green arrows its water accessibility. **c** Motion of six-helix-bundle transporter domains between inward-open (blue) and outward-open states (red), after superposition of the dimerization domains (gray) of the respective average structures. Left: Translation normal to membrane in side view; right: domain rotation (axis indicated by green arrow) seen from above. **d** Projection of the 1-μs equilibrium MD run of the asymmetric dimer onto the plane of vertical displacement $\Delta z_{in}$ and rotation angle $\Delta \phi_{in}$ relative to the inward-open structure (blue: inward-open protomer A; red: outward-open protomer B)

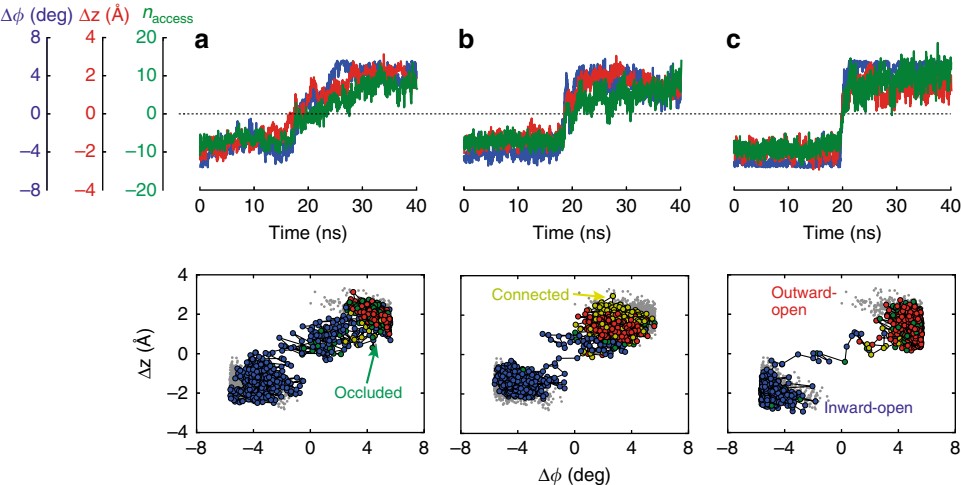

**Fig. 2** Proton- and sodium-transport transition paths. Representative transition paths for **a** H+ transport and **b** Na+ transport at 37 °C, and **c** Na+ transport at 100 °C. The transition-path trajectories are projected onto the two order parameters for rotation and translation of the six-helix-bundle motion, $\Delta \phi$ and $\Delta z$. (Top) Time series of the structural order parameters (blue: $\Delta \phi$; red: $\Delta z$), and the hydration order parameter $n_{access}$ (green). (Bottom) Projection of equilibrium runs (gray) and transition paths onto the $\Delta \phi$ - $\Delta z$ plane. Inward-open, occluded, connected, and outward-open states are shown as blue, green, yellow, and red points, respectively

($n_{access} < 0$). Occluded and connected states have $n_{access} \approx 0$. The paths share a remarkably rapid transition (~10 ns) through the intermediate region, with relaxation processes involving changes in local structure and water access state flanking the actual transition. The transition states (i.e., configurations from which we successfully shot off transition-path trajectories) are intermediate between the metastable states (Supplementary Table 1). The ion-binding site is either occluded (i.e., not connected by water to the solvent layers above and below, or only by fragile

water interactions), or connected to both water layers simultaneously (Supplementary Fig. 10). Note that transiently connected water wires between both sides of the membrane have been observed before in MD simulations of other transporters[22]. A possible concern is that a water spanning pore could result in ion or proton leakage, which would short-circuit the ion exchange. However, with the ion-binding site occupied we did not observe other ions entering. Moreover, the low proton concentration at pH 7 makes it exceedingly unlikely that a proton "arrives" during

a short-lived pore-like intermediate state formed in some transition paths.

Path sampling allowed us to address two key mechanistic questions: whether the transported H$^+$ and Na$^+$ ions take the same path[23,24], and whether temperature alters the transport mechanism. Replacing H$^+$ with Na$^+$ retained the transition-state character (Supplementary Table 1). This conservation for the two branches of the transport cycle strongly suggests that both ions take the same pathway, consistent with the interpretation of the pH-dependent activity[24]. H$^+$ transfer is thus mediated by protonation of Asp159, change of access, and deprotonation. Asp159 also coordinates the transported Na$^+$, together with Glu73 and, via a water molecule, Asp130[3]. Among these three acidic residues, only D159 drastically changes its access state between the inward- and outward-open conformation. Release of the Na$^+$ ions is associated with a flip of the Asp159 side-chain orientation (Supplementary Movie 1), as was proposed in a previous study[7]. By transition-path shooting at 100 °C, we also confirmed that the transition states do not change substantially with temperature (Supplementary Table 1), neither for Na$^+$ nor for H$^+$ transport. However, the transition dynamics is noticeably faster at high temperature, with transition paths spending less time in the barrier region than at 37 °C (Fig. 2c). This speedup is consistent with the rapid increase of the measured transport activity with rising temperature[3].

**Hydrophobic gate**. The transition paths reveal a hydrophobic gate that controls access to the ion-binding site (Fig. 3). The gate is formed by the interlocking residues Ile163 and Tyr255, complemented by Val365 and Leu369. The gate residues are well conserved in the CPA superfamily[4]. Since in the gate one residue is located on the dimerization domain and the other on the transporter domain, gate opening and closing are tightly coupled to domain motion (Fig. 3 and Supplementary Movie 1). The dominant elevator-like 3–4 Å vertical movement of the transporter domain is thus associated with gate opening and closing[20]. In the transition paths, the Ile163-Tyr255 minimum distance increases from ~2.5 to 6 Å as the gate opens (Fig. 3a, b, Supplementary Fig. 11). On the intracellular side, no tightly interacting hydrophobic gate residues were found. Ile69 and Ala132 interact weakly to control solvent access to the ion-binding site (Supplementary Fig. 11). Closing access from the inside also involves motion of helix 5 holding Ala132, a comparably flexible part of the structure[3]. Interestingly, side-chain motions associated with opening and closing of outside and inside access, respectively, exhibited some temporal variability, in particular for H$^+$ transport, likely because the protonated Asp159 interacts less strongly with water than Na$^+$. Importantly, in the transient states with both sides open, the ion-occupied binding site blocks passage of other ions. Conversely, without Na$^+$ or H$^+$ bound, the gate remained closed, which prevents leakage.

A structural alignment of known inward-open conformations of PaNhaP, MjNhaP1, NhaA, and NapA suggests that equivalent residues function as gate in homologous exchangers (Supplementary Table 3 and Supplementary Fig. 12). Thus, the hydrophobic gate between I163 and Y255 appears to be a conserved element of the ion-transport mechanism. In NhaA, residues I168 and T227 interact with F344 and I345 to form the gate (PDB ID: 1ZCD). In the inward-open structure of NapA (PDB ID: 5BZ2), L161 is in contact with V239, and I238 has a contact with I337, forming the gate.

Note that the hydrophobic gate we identified here should be distinguished from the "cytoplasmic gate" reported in a simulation study of NhaA by Beckstein, Shen and collaborators[25]. Their "cytoplasmic gate" is not coupled to the domain motion. It

opens and closes dependent on the protonation state of the proton-carrying aspartate, and remains closed in the inward-open state.

**Ion-exchange activity with mutated hydrophobic gate**. To test the mechanistic predictions of transition-path shooting, we mutated Ile163 and Tyr255 to alanine and measured the ion-exchange activity. The I163A/Y255A mutant (PaNhaP$_{I163A/Y255A}$) with a perturbed hydrophobic gate exchanges ions more than two times faster than wild-type (Fig. 3e, f). This finding is consistent with the expectation that the hydrophobic gate ensures fidelity by minimizing the risk of forming long-lived conducting states, at the expense of overall turnover. To test this hypothesis, we carried out simulations with the I163A/Y255A double mutant in the inward-open conformation. The mutant shows conformational dynamics similar to the wild-type protein, albeit with a weakened gate (Supplementary Fig. 13A), consistent with the experimental data.

To investigate the effect of mutations within the hydrophobic domain interface rather than the gate residues directly, we mutated Val365 and Leu369, both located at the periplasmic part of unwound helix 12 in the helix-bundle, to alanine. In contrast to the I163A/Y255A double mutant, in which each domain carries a single mutation, transport was reduced in comparison to wild-type protein (Supplementary Fig. 14). Since Val365 and Leu369 are located near the ion-binding site and mediate a tight interaction between helices 6 and 12, the mutation might perturb the balance of interactions required for facile domain motion.

**Reaction coordinate for ion exchange**. To improve the efficiency of transition-path shooting simulations[12] and to gain mechanistic insight, we iteratively determined the reaction coordinate $Q$ that describes the transition state of the conformational change. In the optimization, we considered a linear combination of a set of descriptors that contained the angle $\phi$ and $z$-translation of the six-helix-bundle domain and the pair distances resolving opening and closing of the gate (Ile163-Tyr255) and the intracellular residue pair Ile69-Ala132. The likelihood of the outcome of previous shooting attempts can be written in terms of the committor as[12,26,27],

$$\mathcal{L} = \prod_{r_i \to B} \phi_B[Q(r_i)] \prod_{r_j \to A} \left\{ 1 - \phi_B\left[Q\left(r_j\right)\right] \right\}, \quad (1)$$

where A (B) represents the inward (outward) open state, $r_i \to A$ ($r_j \to B$) represents a shooting attempt from initial conformation $r_i$ ($r_j$) that results in the inward (outward) open state, $\phi_B(Q) = [1 + \tanh(Q)]/2$ is the assumed form of the committor function to the outward-open state, and $Q = a_0 + \sum_i a_i q_i$ is the assumed form of the reaction coordinate, with $q_i$ the descriptors listed above and $a_i$ their coefficients. The committor $\phi_B$ is the probability of reaching the outward-open state $B$ first, before the inward-open state $A$, in trajectories initiated from a particular configuration.

We performed a Monte Carlo search in the parameter space $\{a_i\}$ to maximize the likelihood. The resulting coefficients $a_i$ indicate the significance of descriptor $q_i$ in describing the transition state. By using shooting statistics from 16 shooting points of the 2nd round of shooting following the first initial path, we optimized the parameters $\{a_i\}$ (Fig. 4a). For the angle of the domain motion and the distance of the intracellular residue pair Ile69-Ala132, the coefficients take values around zero, which suggests no significant contribution from these order parameters to the transport mechanism. Thus, in the following analysis, we

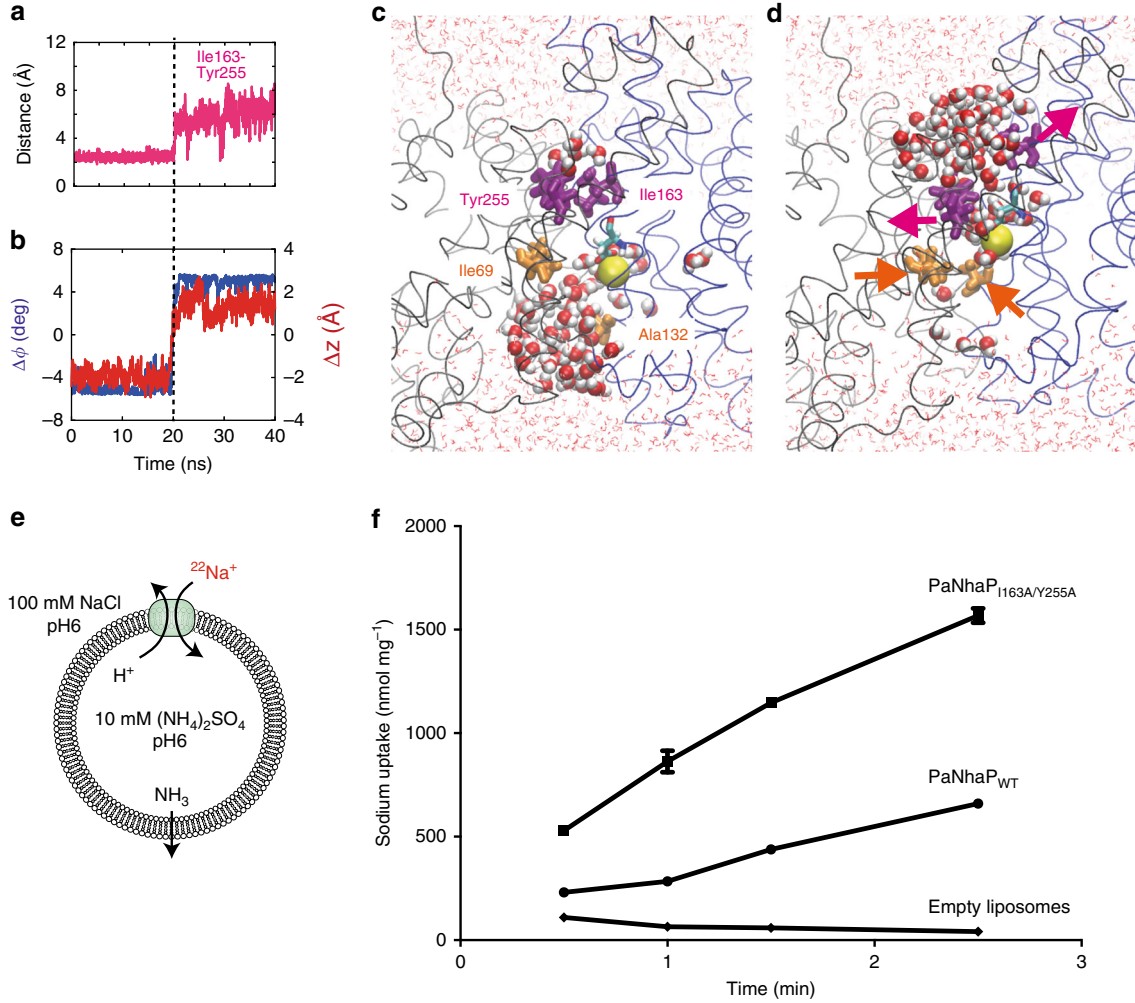

**Fig. 3** Hydrophobic gate. **a** Minimum distances of residue pairs forming the hydrophobic gate (magenta) in transition path at 100 °C with Na$^+$ bound. **b** Concomitant changes of the two structural order parameters for domain motion (angle: blue, left scale; vertical displacement: red, right scale). **c** Structure of inward-open state, with inside access (orange) open and outside access (magenta) closed. Asp159 is shown in stick representation, and the bound Na$^+$ as yellow sphere. Water molecules within 15 Å of the two carboxyl oxygen atoms of Asp159 are shown as spheres; other water molecules are shown as lines, and the protein backbone in gray and blue. **d** Structure of outward-open state. Arrows indicate opening and closing motions. **e** Transport assay of PaNhaP wild type and I163A/Y255A mutant reconstituted in liposomes. Transport was initiated by the dilution of (NH$_4$)$_2$SO$_4$-loaded proteoliposomes into reaction buffer. **f** Efflux of NH$_3$ from the liposomes creates a pH gradient that drives $^{22}$Na uptake by PaNhaP. With 512.6 nmol min$^{-1}$ mg$^{-1}$, the ion exchange activity of the I163A/Y255A mutant is more than two times higher than wild type with 223.5 nmol min$^{-1}$ mg$^{-1}$. Error bars represent the standard error of the mean of ion exchange activity calculated from three independent measurements

only focus on the other two order parameters: the $z$-translation of the moving domain and the distance of the gate residues.

We used the optimized reaction coordinate to select shooting points in the 2nd and 3rd round of path sampling following the second initial path. Conformations that were predicted to have $\phi_B[Q(r_i)] \sim 0.5$ were picked as the 2nd round shooting points. The outcomes of these shots were then added to the likelihood, and the reaction coordinate was optimized again (Fig. 4b). Five shooting points were picked for the 3rd round of shooting attempts based on the optimized reaction coordinate, and the predicted committor $\phi_B[Q(r_i)]$ and observed committor values from the third round of shootings were compared (Fig. 4c). The observed committor values were estimated from 30 shooting trajectories initiated from each shooting point, excluding trajectories that did not commit to either of the two access states during the allotted time (see Methods). Two out of five shooting points show excellent agreement between the predicted and observed committors (shooting conformations 2 and 3 in Fig. 4c). For the other three shooting points, the method underestimated

the committor. However, considering the complexity of the transporter motion and the high dimensionality of the system, we find it remarkable that for four out of five points we indeed obtained some transition paths in a quite small number of trials. One should keep in mind that such transition states are vastly outnumbered by configurations fully committed to either the inward or outward-open states. The shooting conformation 2 in Fig. 4c (II-3r/B in Supplementary Fig. 10 and Supplementary Table 1) that successfully produced transition paths is shown in Fig. 4d. A chain of water molecules connects the buried ion-binding site to the inside and outside surfaces.

The results from the reaction coordinate optimization are consistent with a visual inspection of transition-state conformations (Supplementary Fig. 10). The scatter of the transition states along the $\Delta\phi$ axis suggests that the angle change of the domain motion is not important for describing the transition-state ensemble, consistent with the result from the reaction coordinate optimization. Higher commitment to the outward-open state is correlated with the amount of water access from the outside (or,

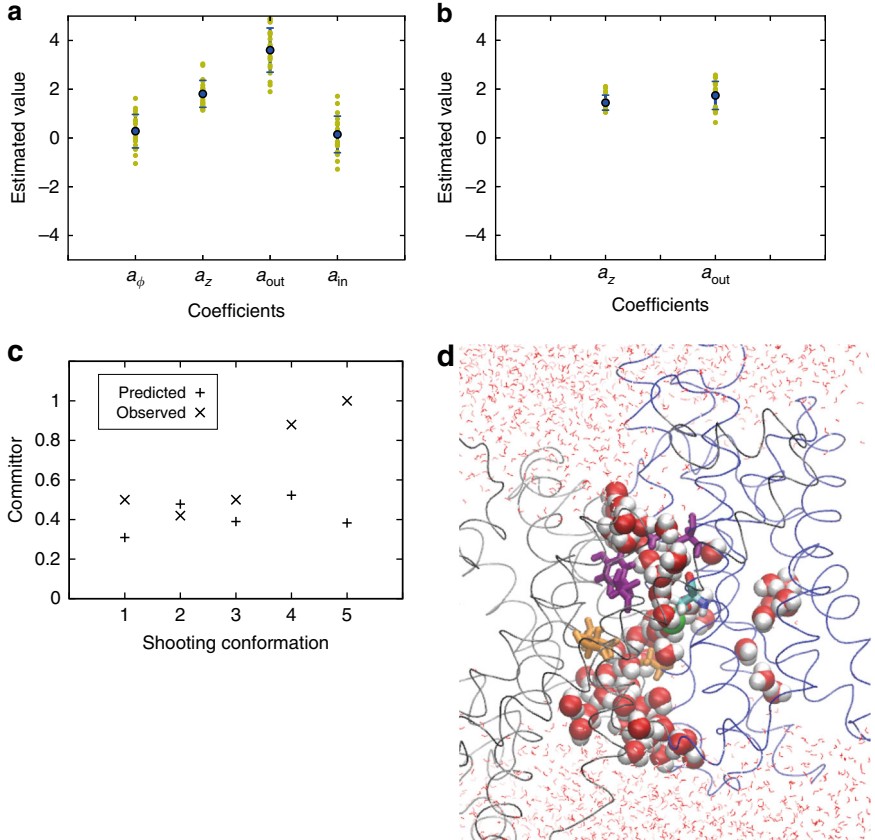

**Fig. 4** Reaction coordinate optimization. **a** Estimated coefficients for four order parameters: angle and z-translation of the domain motion, and pair distances resolving opening and closing of the hydrophobic gate (Ile163-Tyr255) and the intracellular residue pair Ile69-Ala132. The likelihood calculated from shooting data for 16 shooting points was used in the optimization. **b** Estimated coefficients for two order parameters: z-translation of the domain motion and distance of the gate residues. Shooting data from 21 shooting points were used in the likelihood maximization. Error bars in (**a**) and (**b**) represent s.d. of the estimated coefficients. **c** Predicted committor values using the optimized reaction coordinate are compared to the observed committor values from the third round of shootings following the second initial path. **d** A shooting conformation that successfully produced transition paths (shooting conformation 2 in **c**; same graphic representation as in Fig. 3)

equivalently, the opening of the gate); i.e., occluded-state structures with the gate closed have a lower value of the committor.

**Ion-binding and release.** Ion-binding and release events are the remaining steps in a complete transporter cycle (Fig. 5). In our equilibrium simulations of both symmetric and asymmetric dimers, we observed spontaneous $Na^+$ binding and release on the cytosolic and extracellular sides, corresponding to the respective access states (Supplementary Fig. 15). In site B (ion-binding site of protomer B) of the outward-open symmetric dimer, one $Na^+$ stayed at the binding site for ~0.8 μs in a binding mode similar to that in the inward-open conformation, stabilized further by a water bridge to Glu73 (Supplementary Fig. 15C).

**Discussion**
We resolved the unbiased dynamics along the complete ion exchange cycle of PaNhaP, which takes 200 μs at 100 °C or ~ 14 s at room temperature, in atomic detail. The only part of the cycle not accounted for is the quantum-mechanical simulation of (de) protonation of the presumed proton carrier Asp159 in the fully inward-open and outward-open states. However, the timescale reachable in such simulations is at present orders of magnitude shorter than the MD simulations used in this study. Since our main aim is to study the ion shuttling mediated by large-scale

protein conformational changes, thereby overcoming the enormous time-scale gap between seconds-scale ion exchange and microseconds MD simulations, the quantum-mechanical simulation of Asp159 (de)protonation is outside the scope of the present study. The (de)protonation of Asp159 was instead realized by switching between the deprotonated and protonated Asp159 in our MD simulations. The observed ion-binding and structural characteristics in simulations of different protonation states of the residues in the ion-binding site are consistent with Asp159 being responsible for both proton and sodium ion bindings (Supplementary Fig. 16). In our path-sampling simulations of antiporter-mediated ion exchange, both the inward- and outward-open states emerged without any imposed bias, i.e., in MD trajectories evolving freely on a conventional, fully transferable potential energy surface. Remarkably, proton and ion transport requires only small motions of the six-helix-bundle transporter domain (Fig. 1c). The observed 3–4 Å domain translation normal to the membrane and ~10° rotation are about two thirds of the corresponding ~ 6 Å and ~ 15° motions derived from structures of *Thermus thermophilus* NapA (PDB IDs: 5BZ2 and 5BZ3)[7,28], possibly because shuttling two protons simultaneously across the membrane in the electrogenic CPA2-type NapA requires a more pronounced motion. The comparably small scale of the PaNhaP transporter motion should contribute to its high activity of ~5000 ion exchanges per second at the extreme temperatures of *P. abyssi*'s natural environment[3]. In our

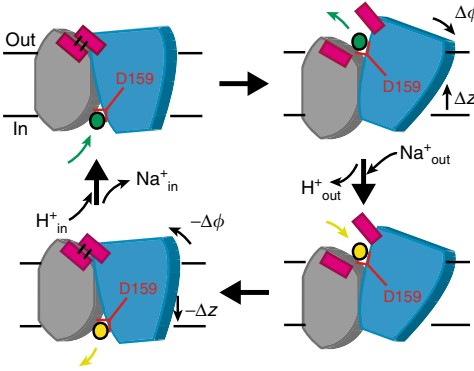

**Fig. 5** Transport cycle. To shuttle ions between the inward-open (left) and outward-open conformations (right), the six-helix-bundle transporter domain (blue) translates along $z$, normal to the membrane, and rotates relative to the core domain (gray) around an axis in the membrane plane. Although the scheme shows the inward transport of Na$^+$ (yellow), and the outward transport of H$^+$ (green), both carried by Asp159, the transport cycle is fully reversible, and its direction determined by the respective gradients of H$^+$ and Na$^+$ across the membrane. Coupled to the domain motion, closing of the hydrophobic gate (magenta) prevents ion leakage

simulations, we could not extract information on the structural origin of the observed pH-sensitive cooperativity in ion transport of PaNhaP[3], which remains unexplained.

Regarding a coupling between ion binding and domain movements, it is important to recognize that ion transport is a rare event. For most of the time, PaNhaP is in a resting state, facing either inwards or outwards, depending on the direction of the last transport event. During these long waiting periods (seconds timescale), ions and protons repeatedly bind and unbind from the ion-binding site (Supplementary Fig. 15). Thus, a spontaneous ion transport event is unlikely to be simulated within the computationally accessible timescale. This limitation was overcome by selectively simulating the rare event with transition path sampling in this study. In our simulations of the relaxed states, we did not observe significant conformational changes triggered directly by ion-binding events or changes in the protonation state, beyond the immediate vicinity of the ion-binding site. Instead, we expect the ion-bound states of the transporter to undergo spontaneous conformational transitions to the respective other access state. These transition paths cross a significant free-energy barrier that is, however, substantially lower than the barrier without bound Na$^+$ or H$^+$ neutralizing the ion-binding site. Indeed, the transition path trajectories show that the conformations "falling off" this barrier from either side quickly relax to conformations close to those seen in unrestrained simulations of inward-facing and outward-facing transporters. Here we did not consider changes in access state without bound ion. Their absence constitutes a key element of the alternating access mechanism of ion exchange. Quantifying the differences in the free-energy barriers and rates of access-state alternation with and without bound ions remains an open challenge.

The competing requirement of ion exchange fidelity is ensured by coupling the global domain motion to the local opening and closing of a gate formed by conserved hydrophobic residues above the ion-binding site. As shown experimentally, perturbing this gate indeed speeds up the transport activity. The strong correlation of gate opening and closing with domain movement and water access to the ion-binding site (Fig. 3a, b), the significant weight of the gate distance in the reaction coordinate (Fig. 4a, b), the tight interactions of the gate residues (Fig. 4b), and the acceleration of transport upon their mutation (Fig. 3f), all point to

residues I163 and Y255 forming a functionally relevant hydrophobic gate for ion passage.

The effect of lipids on the transporter is an important issue. In a previous MD simulation of NapA[29], the lipid bilayer compressed to match the shorter helices of the dimerization domain due to hydrophobic mismatch. We observed a similar effect in our simulation of PaNhaP, albeit with only a modest compression localized to the immediate vicinity of the protein (Supplementary Fig. 17A). Although most of the simulations were performed with a POPE lipid bilayer, an additional simulation was conducted after replacing 20% of the POPE lipids by 1-palmitoyl-2-oleoylphosphatidylglycerol (POPG) to mimic the lipid composition of *E. coli*[28,30], which was used for the transport assay. In ~0.8μs long equilibrium simulations, the presence of POPG lipids did not affect the inward-open and outward-open conformations (Supplementary Fig. 17B).

Our approach of shooting transition paths from the top of the activation barrier extends the arsenal of simulation and modelling techniques of membrane transport[7,19,28,31–34] by probing the dynamics of conformational changes directly, without bias. Transition-path shooting[11,12] can be combined with powerful biased sampling schemes that generate candidate transition paths, such as the procedure developed by Moradi and Tajkhorshid for ABC transporters[35]. As shown here for TMD[16], biased candidate paths provide starting points for transition-path sampling. Transition-path shooting[11,12] should prove useful in mechanistic studies of other molecular machines such as F$_o$F$_1$-ATP synthase, where we expect the transition dynamics between metastable states to proceed likewise on sub-ms timescales[36].

## Methods

**Simulation system setup.** The recently solved ion-bound crystal structure of the PaNhaP dimer at pH8 (PDB ID: 4CZA)[3] was used as initial structure for molecular dynamics (MD) simulations. The dimer was embedded into a 1-palmitoyl-2-oleoylphosphatidylethanolamine (POPE) lipid bilayer and solvated with 150 mM NaCl. The initial box size was $156 \times 126 \times 105$ Å$^3$. Based on PROPKA analysis[37] with nominal pH 7, the buried acidic residues Asp93, Glu359, and Glu408, were protonated. Simulations with different protonation states of the residues Glu73, Asp130, and Asp159 in the ion-binding site, as well as a quantum-mechanical geometry optimization, showed that with Na$^+$ bound, Glu73 has to be protonated and the two aspartates unprotonated to maintain the cation coordination of the crystal structure[3] (Supplementary Fig. 16). These protonation states are further corroborated by the spontaneous binding and release of Na$^+$ ions seen in long equilibrium MD simulations, as essential steps in the functional cycle (Supplementary Fig. 15). One of the paired His292 residues at the dimer interface was protonated to maintain the hydrogen bond interactions. We assume that protonation states are unchanged between 37 and 100 °C[38]. The total number of atoms in the simulation box is ~191,000.

**Molecular dynamics flexible fitting.** After a structural alignment of the inward-open crystal structure of MjNhaP1 (PDB ID: 4CZB) to the outward-open structure (PDB ID: 4D0A) modelled previously on the cryo-EM map[4], we performed flexible fitting by MDFF[15], using a scaling factor ξ = 0.3 kcal/mol. To maintain the secondary structure, harmonic restraints were applied to φ and ψ angles of residues in helices or sheets with a spring constant $k_\mu = 200$ kcal mol$^{-1}$ rad$^{-2}$, and to their backbone hydrogen bonds with a spring constant $k_{\mu'} = 200$ kcal mol$^{-1}$ Å$^{-2}$, with additional restraints ensuring proper peptide stereochemistry. After flexible fitting for 2 ns, we performed 200 steps of energy minimization. During the MDFF run, the cross-correlation coefficient of calculated and measured maps increased from 0.32 to 0.77. The resulting transporter dimer structure was embedded into a 1-palmitoyl-2-oleoylphosphatidylethanolamine (POPE) lipid bilayer and solvated with 150 mM aqueous NaCl solution. The initial box size was $163 \times 130 \times 98$ Å$^3$. The total number of atoms in the simulation box was ~177,000. This system was equilibrated for 68 ns with restraints to the final structure of the MDFF simulation, which were removed in subsequent free MD simulations of 90 ns duration. The backbone of the final structure provided the reference outward-open structure for the TMD simulations.

**Molecular dynamics simulations.** All MD simulations were performed with NAMD2.9/2.10[39] using the CHARMM36 force field[40]. Temperatures of 310 and 373 K were maintained with Langevin dynamics, and the pressure was set to 1 atm (at 310 K) and 200 atm (at 373 K, mimicking the physiological conditions of *Pyrococcus abyssi*) by a Nosé-Hoover Langevin piston[41] with the flexible cell setting.

We used a time step of 2 fs, a real-space cutoff of 12 Å, and particle-mesh Ewald summation for the long-range electrostatics with a mesh width of ~1 Å and cubic interpolation. For the TMD simulations[16], the amino-acid sequences of MjNhaP1 and PaNhaP were aligned by MultiSeq[42]. Cα atoms of the aligned residues were used as target atoms, excluding flexible loop regions. TMD simulations of 10 ns were followed by 50 ns equilibrations with a restraint to the final structure of the TMD simulations, which was removed in the subsequent 1-μs free MD simulations. We performed this procedure twice, with and without symmetrizing the target structure of the TMD simulations. With symmetrisation, both subunits transitioned to an outward-open state; without symmetrisation, only subunit B underwent the transition, completed after ~ 400 ns of equilibrium MD, whereas subunit A relaxed back to an inward-open state. After MD equilibration, the outward-open subunits in the asymmetric ($B_a$) and symmetric ($A_s$ and $B_s$) structures are consistent with each other. The root-mean-square distance (RMSD) of helical Cα-backbone atoms in rigid-body superpositions are 1.3, 1.2, and 1.6 Å for subunit pairs $B_a$-$A_s$, $B_a$-$B_s$, and $A_s$-$B_s$, respectively, using average structures during the last 0.5 μs of MD. For comparison, the corresponding RMSD of subunits A and B averaged over the last 0.5 μs of an equilibrium simulation in an inward-open state starting from the crystal structure is 0.8 Å. For simplicity, we concentrate here on the asymmetric structure in the transition-path shooting simulations. Another TMD simulation of 20 ns from the outward-open conformation after 1-μs equilibration to the inward-open conformation was used as a source of shooting conformations in a second independent transition-path-sampling simulation (second initial path).

**Transition-path shooting**. In the first round of transition-path shooting initiated from the first initial path, starting configurations were obtained from structures at 6, 7, 8, 9 and 10 ns of the 10-ns TMD trajectory towards the outward-open state. From each configuration, five 10-ns trajectories were started with random velocities. Then, in the second round, 16 structures were identified at random from the previous trajectories among the segments that went through the transition regime between the inward- and outward-open states in a projection onto the $\Delta\phi$-$\Delta z$ plane of structural order parameters. From the second round onward, multiple pairs of velocity-inverted trajectories were initiated from each candidate transition state. The initial velocities were drawn randomly according to a Maxwell-Boltzmann distribution for one trajectory segment; for the other segment, sign-inverted velocities were used, such that trajectory segments ending in opposite access states could be smoothly stitched together to form a continuous transition path[12]. Four 40-ns trajectories were run from each configuration, and if they showed at least one successful transition (i.e., two conjugate segments ending up in opposite access states), more trajectory pairs were started from the configuration (Supplementary Table 1).

The path-shooting statistics of rounds 1 and 2 indicated a comparably high percentage of successful shots from structures in which the ion-binding site was either occluded or in hydration contact with both sides of the membrane. In rounds three and four of path shooting, we thus used this access criterion to select candidate structures for transition-path shooting. Importantly, any transport transition path will have a trajectory segment satisfying this condition. As a different and more systematic approach to select candidate shooting points of rounds two and three following the second initial path, we targeted conformations with a committor value ~0.5, as estimated with the help of an iteratively optimized reaction coordinate described as a linear combination of the domain motions and gate distances (see main text). Detailed balance in path sampling defines the relative weights of transition paths found by recursive application of this procedure[12]. The ratio of weights $p_{new}/p_{old}$ of a newly created path and an existing path is accordingly given by the ratio of probabilities $p(old{\rightarrow}new) / p(new{\rightarrow}old)$ to create one path from the other. For our sampling scheme, the ratio of path weights is given by $p_{new}/p_{old} = n_{old} / n_{new}$, i.e., the ratio of the numbers of configurations (saved at uniform time intervals) on the existing and the new path that are in the shooting range[12], which we defined as the union of occluded and connected access states. Here, for simplicity and in reflection of the limited sampling because of large computational costs, we do not weight the paths. For illustrative purposes, the numbers of configurations in the shooting range are shown for 2nd and 3rd rounds of transition path shooting starting from the second initial path (Supplementary Fig. 18A).

Supplementary Table 1 summarizes the transition-path shooting statistics by listing the transition states from which successful transition trajectories were shot off with different ions bound at different temperatures. Also listed are numbers of trajectories reaching either the inward-open or outward-open states, before reaching the other state. Supplementary Table 2 summarizes the computational effort in both equilibrium and path-shooting simulations.

Possible concerns are that the seed path, created by TMD, is quite distorted, crossing the effective barrier late and high, and that the path sampling gives only a limited view of the full transition path ensemble. To address these concerns, we (1) monitored the location of successful transition-path shooting points (which indeed moved toward the center between inward- and outward-facing states; Supplementary Fig. 18B), (2) seeded with a TMD trajectory in the opposite direction (resulting in similar transition pathways after a few rounds of transition path sampling), and (3) checked that the sampling "re-discovered" different mechanistic details (such as changes in the water access states of the ion-binding

sites). Interestingly, intermediates discovered in this way, such as the occluded state, were short lived and did not interfere with the path sampling. Moreover, the pathways projected onto the $\Delta\phi$-$\Delta z$ plane show considerable variability and recurrent behaviour with respect to the detailed route and the dwelling times at intermediate structures (see Supplementary Figs. 6–9). Nevertheless, as with any importance sampling scheme, we cannot entirely exclude the possibility of other relevant pathways, possibly with longer-lived intermediates. Robust features of transition paths were found from the two independent path-sampling simulations initiated from very different initial paths generated by TMD in opposite directions. In the transition paths, the domain motion changes the water-access state of the ion-binding site associated with a flipping motion of Asp159 (Supplementary Movie 1). The outward-open access state tends to emerge at a later stage of the domain motion toward the outward-open state (Supplementary Fig. 19). Considering that the two initial paths have different water-access-state profiles (Supplementary Fig. 19), the convergent behaviour demonstrates that the current method can find reliable and useful paths.

**Order parameters**. The motions of the transporter domains (residues 88–112, 116–140, 145–172, 324–342, 351–377, and 388–411 in PaNhaP) were quantified in terms of rigid-body translation and rotation dynamics. For reference, we used the inward- and outward-open structures obtained by averaging over the corresponding equilibrium MD trajectories during the final 500 ns. First, the protomers were superimposed onto the reference structure using only the dimerization domains (residues 6–27, 31–42, 51–76, 192–224, 231–250, 255–265, and 292–312 in PaNhaP). Then, the translational motions of the transporter domains were obtained by calculating the difference of the target geometric centre from the reference geometric centre of the domain, $\Delta x$, $\Delta y$ and $\Delta z$, relative to the inward- or outward-open structures. The matrix representation of the rotation for optimal rigid-body superposition was used to obtain the angle and axis of the rotational motion. The rotation matrix $R$ of the rotation $\Delta\phi$ along the axis $\vec{e} = (e_1\ e_2\ e_3)^T$ is expressed as,

$$R = (\cos\Delta\phi)I + (1 - \cos\Delta\phi)\,\vec{e}\,\vec{e}^{\,T} + (\sin\Delta\phi)\left[\vec{e}\right]_x, \quad (2)$$

where superscript-T indicates the transpose,

$$\left[\vec{e}\right]_x = \begin{pmatrix} 0 & -e_3 & e_2 \\ e_3 & 0 & -e_1 \\ -e_2 & e_1 & 0 \end{pmatrix} \quad (3)$$

and $I$ is the identity matrix. The trace of $R$ determines the rotation angle,

$$\Delta\phi = \cos^{-1}\left[\frac{1}{2}(R_{11} + R_{22} + R_{33} - 1)\right], \quad (4)$$

which, by definition, we set to be positive. The rotation axis is given by

$$\vec{e} = \left(\frac{R_{32}-R_{23}}{2\sin\Delta\phi}\ \ \frac{R_{13}-R_{31}}{2\sin\Delta\phi}\ \ \frac{R_{21}-R_{12}}{2\sin\Delta\phi}\right). \quad (5)$$

Translation vectors and rotation matrices were determined for optimal superposition with respect to inward-open (subscript in) and outward-open structures (subscript out), and angles and vertical displacement averaged, $\Delta\phi = \left(\Delta\phi_{in} - \Delta\phi_{out}\right)/2$ and $\Delta z = \left(\Delta z_{in} + \Delta z_{out}\right)/2$.

In addition to the structural order parameters, we also quantified water access to the binding site in terms of the following order parameter,

$$n_{access} = \sum_{i\in water} \sigma_r\left(\left|\vec{r}_i - \vec{r}_0\right|\right)\sigma_z(z_i - z_0) \quad (6)$$

$\sigma_r\left(\left|\vec{r}_i - \vec{r}_0\right|\right)$ is a radial sigmoidal function to count only water molecules within some distance $\alpha$ from the centre ($z_i < z_0$) of the ion path, defined as the 3D geometric centre of inside and gate residues,

$$\sigma_r(x) = \frac{1}{1 + \exp[\beta(x-\alpha)]} \quad (7)$$

with $\alpha = 10$ Å, $\beta = 5$. $\sigma_z(z_i-z_0)$ gives negative weights to water molecules on the inward side ($z_i < z_0$) and positive weights to those on the outward side ($z_i > z_0$),

$$\sigma_z(x) = \frac{x}{\gamma}\exp\left(\frac{1}{2} - \frac{x^2}{2\gamma^2}\right), \quad (8)$$

where $\gamma = 5$ Å and $z_0$ is the z coordinate of $\vec{r}_0$.

**Hydration analysis**. To determine the access state of the ion-binding site (inward-open, outward-open, occluded, connected), we analysed the water accessibility to the Oδ atom of residue Asp159, which in our simulations carries the transported proton. The water access state was defined by clustering the water molecules in the system by their pairwise oxygen-oxygen atomic distance, with a cutoff of 3.0 Å. To separate inside and outside water in our 3D periodic system, we excluded a water slab at the centre of the bulk solvent phase parallel to the membrane. We then assigned the water-access state of a particular structure according to the water cluster (or clusters) interacting with the carboxyl Oδ atom of Asp159. In case Oδ interacted with inward (outward) bulk water, we assigned an inward (outward)

water-access state. When Oδ was isolated from both the inward and outward bulk water cluster, it was assigned to the occluded state. In case Oδ was connected to both the inward and outward water clusters (i.e., within 3 Å of at least one water molecule on either side), or if inward and outward water molecules were in one cluster, a connected state was assigned.

Supplementary Fig. 20 shows time series of the resulting water access states, together with illustrative snapshots of the states. The equilibrium trajectory for the asymmetric dimer shows that a fully relaxed outward-open state is established only after about 400 ns of equilibration. This transition at ~400 ns is associated with a slight upward motion (from $\Delta z = 2$–$3$ Å to $\Delta z = 4$ Å) of the transporter domain, and a slight drop in the rotation angle (from $\Delta \phi = 15°$ to $\Delta \phi = 10°$) (Fig. S1B), resulting in a small drop in the RMSD towards the average structure (Supplementary Fig. 1A bottom). For the remaining 600 ns, the chain-A protomer remained in the inward-open state, and the chain-B protomer in the outward-open state (Supplementary Fig. 20A). Note that as a result of the tight 3-Å distance cutoff in water clustering, small water motions suffice to transiently disconnect the ion-binding site from the bulk phase, resulting in assignments to an occluded state. In the symmetric dimer, in which both protomers assumed an outward-open state, water access alternates between outward-open and occluded states in both protomers (Supplementary Fig. 20B), consistent with the access states assigned on the basis of the overall antiporter conformation.

**Transport assay.** PaNhaP$_{WT}$ and the I163A/Y255A mutant (PaNhaP$_{I163A/Y255A}$) were expressed, purified and reconstituted as described here[3]. In brief, PaNhaP was expressed with a cysteine protease domain fused to the C-terminus[43] in E. coli C41-(DE3) cells. Isolated membranes were solubilized in a solubilization buffer containing 20 mM Tris pH 7.4, 150 mM NaCl, 30% Glycerol, 1.5% Cymal-5. Solubilized PaNhaP was bound to Talon resin (Clontech, Mountain View, CA) and eluted through autocleavage of the protease domain induced by 20 μM inositol-hexaphosphate. The transporter was further purified by size exclusion chromatography on a Superdex 200 column equilibrated with 10 mM Na-Citrate pH 4.0, 300 mM NaCl and 0.15 % Cymal-5.

For reconstitution E. coli polar lipid (EPL, Avanti Polar Lipids, Inc., Alabaster, AL) unilamellar vesicles were preformed by extrusion with 400 nm polycarbonate filters. Liposomes were destabilized by adding n-octyl-β-D-glucoside to a final concentration of 1 %. PaNhaP was added to destabilized liposomes and incubated for 1 h at room temperature. Detergent was removed by dialysis overnight. Proteoliposomes were washed and concentrated to a lipid concentration of ~50 mg/ml by ultra-centrifugation before being freshly used in transport assays.

A concentration of 2 μl proteoliposomes with a lipid-to-protein ratio of 100:1 (20 mM Tris/Bis-Tris pH6, 10 mM $(NH_4)_2SO_4$) were diluted in 200 μl reaction buffer (20 mM Tris/Bis-Tris pH6, 10 mM choline chloride, 1 mM MgSO$_4$, 100 μM NaCl, 1μCi/ml $^{22}$Na) to initiate transport. Efflux of NH$_3$ from the liposomes establishes a pH gradient[44], driving uptake of $^{22}$Na by PaNhaP. The protein content of the proteoliposomes was determined by the amido black assay[45].

For the fluorescence assay (Supplementary Fig. 14), 2 μL proteoliposomes with a lipid-to-protein ratio of 100:1 (10 mM Tris/Bis-Tris pH6, 200 mM NaCl) were diluted in 2 mL reaction buffer (10 mM Tris/Bis-Tris pH6, 2 μM acridine orange) to initiate transport. Na$^+$/H$^+$ antiport activity establishes a pH-gradient, which is detected by the quenching of acridine orange fluorescence. Proton gradients were dissipated with 25 mM $(NH_4)_2SO_4$ as control.

**Gate residues from structural alignment.** Gate residues were identified by a structural alignment of the inward-open conformations of PaNhaP, MjNhaP1, NhaA and NapA (PDB IDs: 4CZA, 4CZB, 1ZCD and 5BZ2) by MultiSeq[42] (see Supplementary Table 3). If the PDB file contained more than one chain, chain A was used for the alignment. Since specific local interactions do not affect the structural alignment by MultiSeq, gate residues are occasionally shifted by one residue.

**Reporting summary.** Further information on research design is available in the Nature Research Reporting Summary linked to this article.

## Data availability

The asymmetric dimer conformation of PaNhaP obtained in this study is available in PDB format (Supplementary Data 1). Other data are available from the corresponding authors upon reasonable request.

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

## Acknowledgements

This work was supported by the Max Planck Society, the Frankfurt International Max Planck Research School; and SFB 807 'Transport and communication across biological membranes'. K.O. was supported by Building of Consortia for the Development of Human Resources in Science and Technology, MEXT, Japan, and by JSPS KAKENHI Grant Number JP18H02415. Computations were partially performed using Research Center for Computational Science, Okazaki, Japan. We thank Prof. Klaus Fendler for his comments on the manuscript.

## Author contributions

G.H. and W.K. initiated and supervised the study. K.O., J.W., H.J. and G.H. designed algorithms. K.O., J.W. and H.J. carried out calculations. D.W. and Ö.Y. designed and performed mutagenesis experiments and transport assays. K.O. and G.H. analysed results. K.O. and G.H. wrote the paper (with contributions from W.K.).

## Additional information

**Competing interests:** The authors declare no competing interests.

