## [Peer Review File · Nature Communications]

Reviewers' comments:

Reviewer #1 (Remarks to the Author):

Na⁺/H⁺ exchangers (NHEs) are membrane integrated nanomachines that catalyse the movement of protons and sodium ions across the membrane. The molecular basis of the ion-exchange mechanism is poorly understood. Structures of bacterial homologues are only static crystal structures and dynamic information is required to link these conformational states in order to establish a working energetic landscape of the ion-exchange mechanism. In the manuscript by Hummer and co-workers, sophisticated MD simulations have been used to monitor the structural transitions of the bacterial homologue PaNhaP as it moves from an inward to an outward facing conformation in a membrane. In addition to putting together a richer transport mechanism, they have explored how hydrophobic contacts between the two domains are important and, when modified, can alter transport kinetics.

I think the work is timely and fills an important gap in our current understanding of Na⁺/H⁺ exchangers. I enjoyed reading the paper and I think it is well-written. However, I have some concerns that I think will need to be addressed in order to support the main conclusions of the paper.

Major concerns:

1. Na⁺/H⁺ exchangers are physiological dimers made up of a scaffold domain and a transport domain. The dimer domain helices are shorter than those of the core domain, which results in “bowl” shaped structure, which is particularly prominent in the outward-facing conformation of NapA i.e., similar to that seen in other elevator proteins (Ann. Rev. Biochem. Vol. 85:543-572). In MD simulations, the lipid bilayer compresses to match the shorter dimer domain helices, i.e., due to hydrophobic mismatch (Nat. Comms 13993 (2017)). Furthermore, the transport domains in Na⁺/H⁺ exchangers are tethered to the scaffold domain through very flexible regions (NSMB. 23:248).

Clearly, the lipid bilayer is important component for maintaining the integrity of the Na⁺/H⁺ exchangers and is likely to shape the energetic barriers of the transport mechanism. In MD simulations of PaNhaP, however, the protein has been embedded into only PE lipids. PE is a non-bilayer forming lipid (cone shaped), whereas physiological bilayers also contain bilayer lipids, such as PC or PG. The question is how much of the conclusions from the MD simulations would be different if E. coli membranes rather than PE lipids were used? e.g., one might expect that the elastic properties of the bilayer will be different in bilayers made only from PE lipids and also that the presence of the negatively charged PG lipids would influence the surface charge density. In addition,

its unclear from the current paper if PaNhaP is functional when embedded into liposomes made from only PE lipids as functional data was obtained from liposomes made from E. coli lipids.

Although I appreciate the MD simulations take time to perform, I think it is important to show that similar results are obtained in MD simulations when using bilayers made from a combination of PE:PG lipids. Alternatively, although not as insightful, I think it would be sufficient to show that PaNhaP kinetics were similar when embedded into PE liposomes as compared to E. coli liposomes.

2. The side-chain distances between hydrophobic residues for the proposed “gates” are reported to be 5Å apart. With all due respect, these distances are clearly too large for Van der Waals interactions and so its unclear to me how a “gate” could be formed by them? Indeed, substitution of these residues to alanine in PaNhaP still produced a functional transporter so I dont understand how these residues could form essential “gates” in the transport mechanism.

Is it not also plausible that these hydrophobic surfaces are important for “greasing” the movements of the transport core domains. Modifying these contact regions could change the kinetics as it influences the strength of the contact interactions. Indeed, lipids have been seen in crystal structures of NapA (and a related elevator protein (CitS) to penetrate close to these regions (NSMB. 23:248; eLife 2015;4:e09375).

3. An important consideration is the parameters used to monitor conformational dynamics by MD simulations. Indeed, to replicate the physiological transport process (in addition to using physiological lipids) one would also like to see a coupling between ion-binding and domain movements. Was strict coupling observed between local, ion binding events and larger global domain movements in the MD simulations? For example, if Asp159 was kept in a de-protonated state did one see the same structural transitions from the inward to the outward-state?

Another approach to give confidence in the MD parameters would be to sample the trajectories when a similar non-transported ion was used instead, such as potassium.

Minor concerns

1. Following on from Major comment 2, I think further mutations are required to confirm the importance or exact role of the hydrophobic contact surfaces. At least mutations should be made to the inward-facing hydrophobic residues and also a dramatic mutation (such as substitution to a charged residue) to emphasise the importance of the hydrophobic residues.

2. The term rocker-switch refers too two bundles moving around a centrally-located substrate binding site (Ann. Rev. Biochem. Vol. 85:543-572). In Na⁺/H⁺ exchangers the substrate is instead carried across the membrane by only one of the domains against a scaffold that is fixed, due to oligomerisation. This was first described by Boudker and co-workers as an elevator mechanism. I dont think that the mechanistic insights gained by these studies change this conceptual framework, so I dont agree with the following statement “The elevator-like motion of the transporter domain shuttles the ions back and forth, past the fixed barrier of the dimerization domain; in response to domain motion, the hydrophobic gates open and close to provide the moving barriers of the rocker switch.”

In elevator proteins we still have gating elements. Indeed, in the glutamate transporter the rate-limiting step is the detachment of the transport domain against the scaffold domain (Nature. 2015 Feb 5;518(7537):68-73). These domain interactions are also important for forming cavity-closing contacts, i.e., if they didn't form then the protein would be leaky for ions. From my perspective, the results are consistent with an elevator mechanism. Part of the ambiguity may lie in the use of the term elevator, which is descriptive terminology, and maybe the use of the term “fixed-barrier” mechanism is a conceptually more accurate (Ann. Rev. Biochem. Vol. 85:543-572). Indeed, I think the results from the MD simulations are more accurately defining the extent of movements required to pass the “fixed barrier” as the exchanger transitions from an inward to an outward-facing conformation.

Reviewer #2 (Remarks to the Author):

The manuscript by Kühlbrandt, Hummer and coworkers describes detailed simulations aimed at providing a self consistent mechanism of electroneutral sodium/proton antiporters. As the target for their simulations, the authors use the structure of PaNhaP, an antiporter from hyperthermophilic bacteria. The structure of the protein was obtained facing one side of the membrane, and the authors form the alternate access based on the structure of MjNhaP1. Simulations show that converting the structure of one protein onto the other results in a stable structure. These two structures are the starting points for extensive simulations employing transition-path shooting that are aimed to reproduce the rare event of the transport cycle.

The simulations are clearly performed at the highest possible level, demonstrating how to reach and sample rare events that would be untenable with routine simulations. Moreover, the outcome, i.e. mechanism that the authors propose is very appealing and consistent with other data and models in the literature. Finally, the authors use the simulation results to suggest experiments that are meant

to test the resulting model, and conduct the experiments themselves. This should be the standard for any simulation study.

I do however have a few concerns that I would like the authors to address:

1. Much of the motivation for studying electroneutral sodium/proton antiporters stems from the interest in the mammalian NHE1. Yet there is no detectable sequence similarity between NHE1 and PaNhaP (or MjNhaP1). Obviously, as is the case in other instances, the proteins might be structurally similar, but there is no way of knowing that at this point. Hence, I think that the authors should restrict their discussion at present to PaNhaP and MjNhaP1.

2. As the authors point out, the transport cycle includes protonation and deprotonation of Asp159. Yet the authors did not change the protonation state of this residue during the simulations. Hence, how could the cycle complete itself, or alternatively, how can the simulation mimic the cycle without this essential element?

3. The main key point that is used by the authors to validate their results is the fact that the transport paths reach the correct structures, starting from the alternate access. I am not sure I agree that this is a validation of the path accuracy. The fact that one finds the final destination correctly is by no means a testament that the route is the correct one.

Reviewer #3 (Remarks to the Author):

SUMMARY

The manuscript by Okazaki and colleagues reports on computational work with experimental support to elucidate the conformational transition of the sodium/proton antiporter PaNhaP between inward-facing (IF) and outward-facing (OF) states. The IF state was determined by some of the authors to 3.2 Å resolution by X-ray crystallography in 2014. The OF state is not experimentally known but the authors derive a plausible

structural model for the PaNhaP OF state using an OF model of the related transporter MjNhaP1 (by some of the authors).

KEY RESULTS

The conformational change is primarily a $\sim 5 \text{ \AA}$ rigid body displacement of the transport domain relative to the dimerization domain, together with a small rotation. Sodium ions and protons likely use the same pathway (in contrast to the hypothesis of different pathways stated in the original PaNhaP paper [Ref 6]). At elevated temperature (100°C), the transition pathway closely resembles the pathway at 37°C but the transition is faster, consistent with earlier experimental measurements of the increase of the transport rate with temperature. The authors identify "hydrophobic gates" that additionally control access to the binding site and whose opening appears to be coupled to the "elevator-like" conformational changes. They describe the mechanism as a mixture of the "elevator mechanism" and the "rocking bundle mechanism" of transport, which enhances transporter fidelity.

GENERAL ASSESSMENT

Overall this is an interesting paper that tackles a very important problem in transporter biophysics but is of wider interest as it demonstrates a computational approach that can sample rare events such

as the conformational change of the IF/OF transition and reveal the molecular details.

Combining computation and experiment is excellent but a tighter integration between the two approaches would have been more impactful.

The simulations are clearly very demanding but the authors do not always clearly point out where this might compromise the purported rigor of their approach. There are also some methodological details that should be made clearer.

MAJOR COMMENTS

1) OF model of PaNhaP

Please make the model coordinates available as Supplementary Information or by another means in a public repository/archive with stable access and preferably a DOI (e.g. on figshare, zenodo, DataDryad). This is important so that other researchers could try to reproduce the study or part of the work.

2) Protonation states of PaNhaP

The authors used the pH8 structure of IF PaNhaP with Tl⁺ bound (and replaced it with Na⁺ as far as I understand). Given that PaNhaP

transports a proton, discussion of the protonation states is important.

- Protonation states "were determined by PROPKA" but it is not said at which nominal pH. This needs to be stated clearly.

- pKas can differ at different temperatures, therefore, the protonation states for the simulation at 37°C and 100°C may differ. The authors should demonstrate that their protonation states are consistent and sensible.

- Show the data that support the assignment of protonation states of D159, D130, and E73 in the binding site

"Simulations with different protonation states of the residues Glu73, Asp130, and Asp159 in the ion binding site, as well as a quantum-mechanical geometry optimization, showed that with Na⁺ bound, Glu73 has to be protonated and the two aspartates unprotonated to maintain the cation coordination of the crystal structure."

(This is only stated without further evidence.)

- What is the evidence that D159 is the proton carrying residue?

3) Shooting-simulations

The "shooting from the top" method was introduced by some of the authors in [Jung et al JCP 147 (2017) 152716] in a rigorous fashion. Seeing it applied to a real macromolecular system is exciting. However, it seems that some of the rigor was sacrificed and this is not made very clear.

As far as I can tell, in this paper the authors do not perform a Metropolis move in order to decide to accept a newly generated transition path (TP), mainly because they only generate so few full TPs in the first place. However, this implies that the generated paths are not guaranteed to be members of the equilibrium TP ensemble, or more precisely, their true weights might be so small as to be non-representative of "typical" equilibrium transitions?

The initial starting points are generated from TMD, which can distort proteins fairly strongly. It is thus quite possible that these conformations in the transition region were very rare if they were drawn from paths in the real TP ensemble and thus the sampled TPs might not be true "equilibrium paths". I.e., whatever is seen in the simulations might still be heavily influenced by the initial TMD. Perhaps the reason that the "barrier crossing" time is short is because the intermediates are somewhat non-physical or unlikely "excited states"?

The claim (abstract) that the simulations cover the entire transport cycle with "continuous and unbiased molecular dynamics trajectories" is therefore not quite true, or at least requires a

more nuanced discussion (instead of implying that the rigorous JCP paper fully justifies the approach here). The sentence (line 95)

"These transition paths are proper representatives of the ensemble of transition paths that would be obtained in an infinitely long simulation."

seems misleading, given that it implies that the generate TPs are typical or highly likely to be observed.

Please discuss the potential short-comings of the TP ensemble as generated here and preferably show evidence of how far generated paths are away from equilibrium.

4) Connection between simulations and experiments

The functional experiments for the I163A/Y255A double mutant are somewhat disconnected from the simulations and are used in a hand-waving manner to ascribe a function to the outer hydrophobic gate, namely to ensure fidelity at the expense of efficiency.

I suggest the authors should test the same mutant in simulations and compare to experiment.

MINOR COMMENTS

1) Modeling of an alternative conformation of a transporter by a procedure similar to the one used here was done (to my knowledge) for the first time by (using TMD towards another transporter)

S. A. Shaikh and E. Tajkhorshid. Modeling and dynamics of the inward-facing state of a Na⁺/Cl⁻ dependent neurotransmitter transporter homologue. *PLoS Comput Biol*, 6(8):e1000905, 2010.

and more recently (using MDFF directly) by

N. Coudray, S. Seyler, R. Lasala, Z. Zhang, K. M. Clark, M. E. Dumont, A. Rohou, O. Beckstein, and D. L. Stokes. Structure of the SLC4 transporter Bor1p in an inward-facing conformation. *Protein Sci*, 26(1):130–145, 2017.

I would consider citing Shaikh et al; the second reference is just to say that this computational protocol seems to become common.

2) Please comment on the need to use MDFF with the cryo-electron crystallography density for the OF MjNhaP1 model. Why did the authors not simply use the published model with PDB ID 4D0A? How does their OF model differ from 4D0A?

(Given the importance of the target conformation in the modeling procedure one would like to know what the features in the OF MjNhaP1 density are that ultimately determine the conformation of the PaNhaP structure.)

3) Some of the transition paths show that the ion binding site is sometimes connected to both compartments. Is this an indication of a leak for protons or Na⁺? --- water pathways have been observed for other transporters, see e.g.

J. Li, S. A. Shaikh, G. Enkavi, P.-C. Wen, Z. Huang, and E. Tajkhorshid. Transient formation of water-conducting states in membrane transporters. *Proc Natl Acad Sci U S A*, 110(19):7696–7701, Apr 2013.

so that wouldn't worry me. But what is the evidence that these water-wires could not transport protons?

4) Experiments by some of the authors [Wöhlert et al *eLife* (2014)] show that PaNhaP transport is cooperative at pH6 but not at pH5. This was taken as a major point in order to propose PaNhaP as a good model for human NHEs.

The experiments in this paper were performed at pH6. As mentioned above, it is not clear at which nominal pH the simulations are supposed to occur but assuming that protonation states were chosen

to be compatible with the experiments, cooperativity would be expected in the transport simulations. Namely, it should make a difference if both protomers are moving in parallel or if only one moves or if they moved in an anti-parallel fashion.

Please comment and at least make clear to the reader in how far the simulations address the remarkable experimental finding of cooperative transport.

5) Can the authors use any of their data to approximate an underlying free energy landscape and a reaction-coordinate dependent diffusion coefficient?

(Or is this impossible because the weights of the TP in TP ensemble are not known?)

Response to Reviewers' comments

Reviewers' comments are shown in *Italic*.

Reviewer #1

I think the work is timely and fills an important gap in our current understanding of Na⁺/H⁺ exchangers. I enjoyed reading the paper and I think it is well-written. However, I have some concerns that I think will need to be addressed in order to support the main conclusions of the paper.

We thank the reviewer for the encouraging comment.

1. Na⁺/H⁺ exchangers are physiological dimers made up of a scaffold domain and a transport domain. The dimer domain helices are shorter than those of the core domain, which results in "bowl" shaped structure, which is particularly prominent in the outward-facing conformation of NapA I.e., similar to that seen in other elevator proteins (Ann. Rev. Biochem. Vol. 85:543-572). In MD simulations, the lipid bilayer compresses to match the shorter dimer domain helices, I.e., due to hydrophobic mismatch (Nat. Comms 13993 (2017)). Furthermore, the transport domains in Na⁺/H⁺ exchangers are tethered to the scaffold domain through very flexible regions (NSMB. 23:248).

In response to this interesting suggestion, we examined the local compression of the bilayer. In the newly added Supplementary Fig. 16A, we show that the bilayer thickness is indeed reduced. However, the compression is quite modest and localized to the immediate vicinity of the protein.

Clearly, the lipid bilayer is important component for maintaining the integrity of the Na⁺/H⁺ exchangers and is likely to shape the energetic barriers of the transport mechanism. In MD simulations of PaNhaP, however, the protein has been embedded into only PE lipids. PE is a non-bilayer forming lipid (cone shaped), whereas physiological bilayers also contain bilayer lipids, such as PC or PG. The question is how much of the conclusions from the MD simulations would be different if E. coli membranes rather than PE lipids were used? e.g., one might expect

that the elastic properties of the bilayer will be different in bilayers made only from PE lipids and also that the presence of the negatively charged PG lipids would influence the surface charge density. In addition, its unclear from the current paper if PaNhaP is functional when embedded into liposomes made from only PE lipids as functional data was obtained from liposomes made from E. coli lipids.

Although I appreciate the MD simulations take time to perform, I think it is important to show that similar results are obtained in MD simulations when using bilayers made from a combination of PE:PG lipids. Alternatively, although not as insightful, I think it would be sufficient to show that PaNhaP kinetics were similar when embedded into PE liposomes as compared to E. coli liposomes.

We addressed this concern by simulating PaNhaP also in POPE:POPG mixed lipid membrane. Starting from the asymmetric dimer conformation, no significant structural changes were observed in the simulation, neither for the inward-facing nor for the outward-facing monomer. See Supplementary Fig. 16B.

We now write in the Discussion: “The effect of lipids on the transporter is an important issue. In a previous MD simulation of NapA, the lipid bilayer compressed to match the shorter helices of the dimerization domain due to hydrophobic mismatch. We observed a similar effect in our simulation of PaNhaP, albeit with only a modest compression localized to the immediate vicinity of the protein (Supplementary Fig. 16A). Although most of the simulations were performed with a POPE lipid bilayer, an additional simulation was conducted after replacing 20% of the lipids by 1-palmitoyl-2-oleoylphosphatidylglycerol (POPG) to mimic the lipid composition of E. coli, which was used for the transport assay. In ~0.8 μ s long equilibrium simulations, the presence of POPG lipids did not affect the inward-open and outward-open conformations (Supplementary Fig. 16B).”

2. The side-chain distances between hydrophobic residues for the proposed “gates” are reported to be 5Å apart. With all due respect, these distances are clearly too large for Van der Waals interactions and so its unclear to me how a “gate” could be formed by them? Indeed, substitution of these residues to alanine in PaNhaP still produced a functional transporter so I dont understand how these residues could form essential “gates” in the transport mechanism.

We apologize for not clearly having communicated that we had reported distances between side-chain centers of mass. To avoid further confusion, we now report the minimum distance between the gate residues, which indeed drops below $\sim 3 \text{ \AA}$ for closed gates (see Supplementary Fig. 11). We modified text in Results accordingly: “In the transition paths, the Ile163-Tyr255 minimum distance increases from ~ 2.5 to 6 \AA as the outside gate opens. Concomitantly, the Ile69-Ala132 minimum distance shrinks from 8 to 3 \AA as the inside gate closes (Fig. 3A-B, Supplementary Fig. 11).”

Is it not also plausible that these hydrophobic surfaces are important for “greasing” the movements of the transport core domains. Modifying these contact regions could change the kinetics as it influences the strength of the contact interactions. Indeed, lipids have been seen in crystal structures of NapA (and a related elevator protein (CitS) to penetrate close to these regions (NSMB. 23:248; eLife 2015;4:e09375).

We thank the Reviewer for this interesting suggestion. In further support of our finding for a functional role of the hydrophobic residues in gating access to the ion binding site, we now point more clearly to the fact that in the optimization of the reaction coordinate (Fig. 4A-B), the opening of the outward gate (I163-Y255) emerged as a major component of the transport mechanism. We now also clarify that the inward gate (I69-A132) does not weigh heavily in the optimized reaction coordinate. These findings are consistent with the weaker interaction of the inward-gate residues, as shown in Supplementary Fig. 11. Putting these observations together we now state in Discussion: “The strong correlation of gate opening and closing with domain movement and water access to the ion binding site (Fig. 3A-B), the significant weight of the outward-gate distance in the reaction coordinate (Fig. 4A-B), the tight interactions of the outward-gate residues (Fig. 4B), and the acceleration of transport upon their mutation (Fig. 3F), all point to residues I163 and Y255 forming a functionally relevant hydrophobic gate for ion passage. For the inner-gate residues I69 and A132, additional functions may be important, e.g., to “grease” the movements of the transport core.”

3. An important consideration is the parameters used to monitor conformational dynamics by MD simulations. Indeed, to replicate the physiological transport process (in addition to using physiological lipids) one would also like to see a coupling between ion-binding and domain

movements. Was strict coupling observed between local, ion binding events and larger global domain movements in the MD simulations?

In response to this question, we now write in the Discussion: “Regarding a coupling between ion binding and domain movements, it is important to recognize that ion transport is a rare event. For most of the time, PaNhaP is in a resting state, facing either inwards or outwards, depending on the direction of the last transport event. During these long waiting periods, ions and protons repeatedly bind and unbind from the ion-binding site (Supplementary Fig. 15). In our simulations of the relaxed states, we did not observe significant conformational changes triggered directly by ion binding events or changes in the protonation state, beyond the immediate vicinity of the ion-binding site. Instead, we expect the ion-bound states of the transporter to undergo spontaneous conformational transitions to the respective other access state. These transition paths cross a significant free-energy barrier that is, however, substantially lower than the barrier without bound Na⁺ or H⁺ neutralizing the ion-binding site. Indeed, the transition path trajectories show that the conformations “falling off” this barrier from either side quickly relax to conformations close to those seen in unrestrained simulations of inward-facing and outward-facing transporters.”

For example, if Asp159 was kept in a de-protonated state did one see the same structural transitions from the inward to the outward-state?

We have not performed transition path sampling simulations of the passage of the transport core domain without bound ion because there is no experimental evidence for uncoupled transitions (Woehlert et al, eLife 2014, Calinescu et al., JBC 2016) under physiological conditions. Nevertheless, we recognize that this absence of uncoupled passages is mechanistically interesting. However, it is highly challenging to quantify the strength of the coupling of transporter-core passage to the ion binding state because it requires very precise calculations of the free energy barrier for a large and complex conformational change. Such calculations with an accuracy of <1 kT are beyond reach. In Discussion we now write: “Here we did not consider changes in access state without bound ion. Their absence constitutes a key element of the alternating access mechanism of ion exchange. Quantifying the differences in the free-energy barriers and rates of access-state alternation with and without bound ions remains an open challenge.”

Another approach to give confidence in the MD parameters would be to sample the trajectories when a similar non-transported ion was used instead, such as potassium.

We have studied potassium binding and are in the process of completing a separate manuscript on this. In short, we find that potassium binding is indeed much weaker than sodium binding. However, once bound, K^+ ions are accommodated in the binding site without substantial differences. On the basis of these preliminary results, we tend to conclude that K^+ is not transported because it does bind competitively with Na^+ , not because it could not be shuttled by the transporter core. Therefore, transition path sampling would not shed light on this interesting question.

Minor concerns

1. Following on from Major comment 2, I think further mutations are required to confirm the importance or exact role of the hydrophobic contact surfaces. At least mutations should be made to the inward-facing hydrophobic residues and also a dramatic mutation (such as substitution to a charged residue) to emphasise the importance of the hydrophobic residues.

We thank the Reviewer for this suggestion. In response, we now make it clearer, based on our reaction coordinate optimization (Fig. 4), that the inward-facing hydrophobic gate I69-A132 is likely a less important part of the mechanism. We also make it clear that the I69-A132 interaction is weaker than the I163-Y255 interaction in the respective closed gates. In light of this reduced role and the close proximity of these residues to the ion binding site, we did not perform mutations at this site. However, we now have performed simulations of an I69A mutant, with both residues of the inward-facing hydrophobic gate being alanines (Supplementary Figures 13) and created the PaNhaP double mutant V365A-L369A experimentally to further investigate the role of the outside gate and the influence of the hydrophobic domain interface (Supplementary Figures 14). Since the distance of the outside gate residues to the ion binding site is small ($\sim 6.5\text{\AA}$) we did not introduce charged residues to exclude a potential effect on ion binding rather than gating.

Now we write in Results: “To investigate the effect of mutations within the hydrophobic domain interface rather than the gate residues directly, we mutated Val365 and Leu369, both located at the periplasmic part of unwound helix 12 in the helix-bundle, to alanine. In contrast to the I163A/Y255A double mutant, in which each domain carries a single mutation, transport was reduced in comparison to wildtype protein (Supplementary Fig. 14). Since Val365 and Leu369 are located near the ion-binding site and mediate a tight interaction between helices 6 and 12, the mutation might perturb the balance of interactions required for facile domain motion. We also performed simulations of the inside-gate mutant I69A in an outward-open conformation (Supplementary Fig. 13B). In this mutant, the inside-gate becomes leaky, transiently producing connected water wires.”

2. The term rocker-switch refers too two bundles moving around a centrally-located substrate binding site (Ann. Rev. Biochem. Vol. 85:543-572). In Na⁺/H⁺ exchangers the substrate is instead carried across the membrane by only one of the domains against a scaffold that is fixed, due to oligomerisation. This was first described by Boudker and co-workers as an elevator mechanism. I dont think that the mechanistic insights gained by these studies change this conceptual framework, so I dont agree with the following statement “The elevator-like motion of the transporter domain shuttles the ions back and forth, past the fixed barrier of the dimerization domain; in response to domain motion, the hydrophobic gates open and close to provide the moving barriers of the rocker switch.”

In elevator proteins we still have gating elements. Indeed, in the glutamate transporter the rate-limiting step is the detachment of the transport domain against the scaffold domain (Nature. 2015 Feb 5;518(7537):68-73). These domain interactions are also important for forming cavity-closing contacts, I.e., if they didn't form then the protein would be leaky for ions. From my perspective, the results are consistent with an elevator mechanism. Part of the ambiguity may lie in the use of the term elevator, which is descriptive terminology, and maybe the use of the term “fixed-barrier” mechanism is a conceptually more accurate (Ann. Rev. Biochem. Vol. 85:543-572). Indeed, I think the results from the MD simulations are more accurately defining the extent of movements required to pass the “fixed barrier” as the exchanger transitions from an inward to an outward-facing conformation.

In response, we have rephrased the statement: “The **dominant** elevator-like motion of the transporter domain shuttles the ions back and forth, past the fixed barrier of the dimerization

domain; associated with this 3-4 Å vertical movement are, as elements of a rocker switch, a 10° domain rotation and an opening and closing of the hydrophobic gates above and below the moving ion-binding site.”

Reviewer #2

The manuscript by Kühlbrandt, Hummer and coworkers describes detailed simulations aimed at providing a self consistent mechanism of electroneutral sodium/proton antiporters. As the target for their simulations, the authors use the structure of PaNhaP, an antiporter from hyperthermophilic bacteria. The structure of the protein was obtained facing one side of the membrane, and the authors form the alternate access based on the structure of MjNhaP1. Simulations show that converting the structure of one protein onto the other results in a stable structure. These two structures are the starting points for extensive simulations employing transition-path shooting that are aimed to reproduce the rare event of the transport cycle. The simulations are clearly performed at the highest possible level, demonstrating how to reach and sample rare events that would be untenable with routine simulations. Moreover, the outcome, i.e. mechanism that the authors propose is very appealing and consistent with other data and models in the literature. Finally, the authors use the simulation results to suggest experiments that are meant to test the resulting model, and conduct the experiments themselves. This should be the standard for any simulation study.

We thank the reviewer for such encouraging comments.

I do however have a few concerns that I would like the authors to address:

1. Much of the motivation for studying electroneutral sodium/proton antiporters stems from the interest in the mammalian NHE1. Yet there is no detectable sequence similarity between NHE1 and PaNhaP (or MjNhaP1). Obviously, as is the case in other instances, the proteins might be structurally similar, but there is no way of knowing that at this point. Hence, I think that the authors should restrict their discussion at present to PaNhaP and MjNhaP1.

In response, we refer to the earlier publications [Goswami et al. EMBO J 2011] and [Paulino et al. eLife 2014], in which a significant sequence similarity between NHE1 and PaNhaP (or MjNhaP1) was established. Sequence alignment in Figure 3 of [Goswami et al. EMBO J 2011] or “Figure 2 – figure supplement 4” of [Paulino et al. eLife 2014] shows a strong conservation of binding site residues on helices 5 and 6. Therefore, we think it is appropriate to discuss a common mechanism of these homologous transporters. We now include this information in the Introduction: “The pronounced sequence homology of archaeal and mammalian ion exchangers, especially in the cation binding site justifies the use of PaNhaP and MjNhaP1 as model systems in mechanistic studies of electroneutral Na⁺/H⁺ exchange.”

2. As the authors point out, the transport cycle includes protonation and deprotonation of Asp159. Yet the authors did not change the protonation state of this residue during the simulations. Hence, how could the cycle complete itself, or alternatively, how can the simulation mimic the cycle without this essential element?

We recognize that the protonation and deprotonation of D159 are important parts of the cycle. However, (de)protonation events are inherently quantum mechanical. To simulate these processes directly, one would need to perform a quantum mechanical simulation that is costly and out of scope here. In our paper, we stress this issue by writing: “The only part of the cycle not accounted for is the quantum-mechanical simulation of (de)protonation of the proton carrier Asp159.” However, we performed some (classical) MD simulations in which the protonation state of D159 was changed (Supplementary Fig. 17). These simulations show the importance of D159 in ion transports by PaNhaP. Our simulations indicate that deprotonated D159 has a strong affinity to cations. Rough pKa estimates (see response to Reviewer #3 below) give a pKa value of 11-12, indicating a competing strong affinity for protons.

3. The main key point that is used by the authors to validate their results is the fact that the transport paths reach the correct structures, starting from the alternate access. I am not sure I agree that this is a validation of the path accuracy. The fact that one finds the final destination correctly is by no means a testament that the route is the correct one.

To clarify this point, spontaneously reaching inward-facing and outward-facing conformations from a common starting point in phase space is a fundamental requirement of transition path

sampling by shooting, not a validation. Every such trajectory then constitutes, by construction, a valid transition path. The open question is how typical any such path is of the entire ensemble of transition path. As in any importance sampling method (such as molecular dynamics and Monte Carlo simulations), one gains confidence, first, by initiating the search from different seed pathways and, second, by repeating the sampling procedure and comparing newly found pathways to previous pathways. As we show in the paper, we obtained consistent results by seeding the search with a targeted-MD path in the opposite direction, where one would expect a very different bias a priori. We also show that our transition path sampling produces pathways which maintain their general character (in particular with regards to the transporter core motion), yet vary in other aspects (in particular with regards to the hydration states and transient intermediates such as an occluded state). On the basis of these findings, we conclude that we could collect a small but representative sample of the transition path ensemble.

We now write in Methods: “Possible concerns are that the seed path, created by TMD, is quite distorted, crossing the effective barrier late and high, and that the path sampling gives only a limited view of the full transition path ensemble. To address these concerns, we (1) monitored the location of successful transition-path shooting points (which indeed moved toward the center between inward and outward facing states; Supplementary Fig. 18B), (2) seeded with a TMD trajectory in the opposite direction (resulting in similar transition pathways after a few rounds of transition path sampling), and (3) checked that the sampling “re-discovered” different mechanistic details (such as changes in the water access states of the ion binding sites). Interestingly, intermediates discovered in this way, such as the occluded state, were short lived and did not interfere with the path sampling. Moreover, the pathways projected onto the $\Delta\phi$ - Δz plane show considerable variability and recurrent behavior with respect to the detailed route and the dwelling times at intermediate structures (see Supplementary Figs. 6-9). Nevertheless, as with any importance sampling scheme, we cannot entirely exclude the possibility of other relevant pathways, possibly with longer-lived intermediates.”

Reviewer #3

SUMMARY

The manuscript by Okazaki and colleagues reports on computational work with experimental support to elucidate the conformational transition of the sodium/proton antiporter PaNhaP between inward-facing (IF) and outward-facing (OF) states. The IF state was determined by some of the authors to 3.2 Å resolution by X-ray crystallography in 2014. The OF state is not experimentally known but the authors derive a plausible structural model for the PaNhaP OF state using an OF model of the related transporter MjNhaP1 (by some of the authors).

KEY RESULTS

The conformational change is primarily a ~5 Å rigid body displacement of the transport domain relative to the dimerization domain, together with a small rotation. Sodium ions and protons likely use the same pathway (in contrast to the hypothesis of different pathways stated in the original PaNhaP paper [Ref 6]). At elevated temperature (100°C), the transition pathway closely resembles the pathway at 37°C but the transition is faster, consistent with earlier experimental measurements of the increase of the transport rate with temperature. The authors identify "hydrophobic gates" that additionally control access to the binding site and whose opening appears to be coupled to the "elevator-like" conformational changes. They describe the mechanism as a mixture of the "elevator mechanism" and the "rocking bundle mechanism" of transport, which enhances transporter fidelity.

GENERAL ASSESSMENT

Overall this is an interesting paper that tackles a very important problem in transporter biophysics but is of wider interest as it demonstrates a computational approach that can sample rare events such as the conformational change of the IF/OF transition and reveal the molecular details.

Combining computation and experiment is excellent but a tighter integration between the two approaches would have been more impactful.

The simulations are clearly very demanding but the authors do not always clearly point out where this might compromise the purported rigor of their approach. There are also some methodological details that should be made clearer.

We thank the Reviewer for these positive and encouraging comments. In response, we now emphasize more clearly the limitations of our approach. In particular, we make it clear that we could collect only a small number of transition pathways (which, however, do allow us to identify commonalities and differences), and that with the limited data we could not calculate free energies and rates for the access change dynamics.

MAJOR COMMENTS

1) OF model of PaNhaP

Please make the model coordinates available as Supplementary Information or by another means in a public repository/archive with stable access and preferably a DOI (e.g. on figshare, zenodo, DataDryad). This is important so that other researchers could try to reproduce the study or part of the work.

We now include a PDB file of the relaxed structure as supplementary information (Supplementary Data 1).

2) Protonation states of PaNhaP

The authors used the pH8 structure of IF PaNhaP with Tl⁺ bound (and replaced it with Na⁺ as far as I understand). Given that PaNhaP transports a proton, discussion of the protonation states is important.

Protonation states "were determined by PROPKA" but it is not said at which nominal pH. This needs to be stated clearly.

We now state that the protonation states correspond to pH 7 in Methods: "Based on PROPKA analysis with nominal pH7, ...".

- pKas can differ at different temperatures, therefore, the protonation states for the simulation at 37°C and 100°C may differ. The authors should demonstrate that their protonation states are consistent and sensible.

The Reviewer raises an interesting point. A literature study [Journal of Chromatography A, 1155 (2007) 142–145] indicates that for carboxylic acids dpKa/dT values are very small and the

pKa values change less than ± 0.05 units/10 K. Therefore, we assume that protonation states are basically not changed between 37°C and 100°C. We now state this assumption in Methods: “We assume that protonation states are unchanged between 37°C and 100°C.”

- Show the data that support the assignment of protonation states of D159, D130, and E73 in the binding site "Simulations with different protonation states of the residues Glu73, Asp130, and Asp159 in the ion binding site, as well as a quantum-mechanical geometry optimization, showed that with Na⁺ bound, Glu73 has to be protonated and the two aspartates unprotonated to maintain the cation coordination of the crystal structure." (This is only stated without further evidence.)

The pKa values of E73, D130, and D159 were estimated as 8, 5~6 and 11~12 (in the absence of Na⁺), respectively, by PROPKA. Thus, D130 was kept deprotonated, while protonation states of E73 and D159 were explored (Supplementary Fig. 17). The simulation results indicate that, E73 is protonated, while D159 is either deprotonated with Na⁺ bound or protonated. The high pKa of D159 in the absence of sodium is likely relevant for function, ensuring that protons (at their very low concentration at near-neutral pH) can compete with abundant Na⁺.

- What is the evidence that D159 is the proton carrying residue?

We now state in Results section that “Among these three acidic residues, only D159 drastically changes its access state between the inward- and outward-open conformation”, making it very likely to be the proton and ion carrier. In addition, sequence comparison ([Goswami et al. EMBO J 2011], [Paulino et al. eLife 2014]) suggests that D159 is the most conserved residue in the binding site of CPA1 (ND motif) as well as CPA2 (DD motif) antiporters, further emphasizing its importance for transport.

3) Shooting-simulations

The "shooting from the top" method was introduced by some of the authors in [Jung et al JCP 147 (2017) 152716] in a rigorous fashion. Seeing it applied to a real macromolecular system is exciting. However, it seems that some of the rigor was sacrificed and this is not made very clear. As far as I can tell, in this paper the authors do not perform a Metropolis move in order to decide to accept a newly generated transition path (TP), mainly because they only generate so few full

TPs in the first place. However, this implies that the generated paths are not guaranteed to be members of the equilibrium TP ensemble, or more precisely, their true weights might be so small as to be non-representative of "typical" equilibrium transitions?

In response, we now clarify that with the few rounds of iteration performed with a proper "shooting range" (as defined by Jung et al.), the effect of reweighting is minimal (Supplementary Fig. 18A). The reason is that the relative weight factors are typically in the range of 0.1 to 1, as given by the relative times spent in the shooting range. Only after many rounds, the repeated application of such factors would lead to large distortions. We now write in Methods: "For illustrative purposes, the numbers of configurations in the shooting range are shown for 2nd and 3rd rounds of transition path shooting starting from the second initial path (Supplementary Fig. 18A)."

The initial starting points are generated from TMD, which can distort proteins fairly strongly. It is thus quite possible that these conformations in the transition region were very rare if they were drawn from paths in the real TP ensemble and thus the sampled TPs might not be true "equilibrium paths". I.e., whatever is seen in the simulations might still be heavily influenced by the initial TMD. Perhaps the reason that the "barrier crossing" time is short is because the intermediates are somewhat non-physical or unlikely "excited states"?

In response, we now write in Methods: "Possible concerns are that the seed path, created by TMD, is quite distorted, crossing the effective barrier late and high, and that the path sampling gives only a limited view of the full transition path ensemble. To address these concerns, we (1) monitored the location of successful transition-path shooting points (which indeed moved toward the center between inward and outward facing states; Supplementary Fig. 18B), (2) seeded with a TMD trajectory in the opposite direction (resulting in similar transition pathways after a few rounds of transition path sampling), and (3) checked that the sampling "re-discovered" different mechanistic details (such as changes in the water access states of the ion binding sites). Interestingly, intermediates discovered in this way, such as the occluded state, were short lived and did not interfere with the path sampling. Moreover, the pathways projected onto the $\Delta\phi$ - Δz plane show considerable variability and recurrent behavior with respect to the detailed route and the dwelling times at intermediate structures (see Extended Data Figures 6-9).

Nevertheless, as with any importance sampling scheme, we cannot entirely exclude the possibility of other relevant pathways, possibly with longer-lived intermediates.”

The claim (abstract) that the simulations cover the entire transport cycle with "continuous and unbiased molecular dynamics trajectories" is therefore not quite true, or at least requires a more nuanced discussion (instead of implying that the rigorous JCP paper fully justifies the approach here). The sentence (line 95) "These transition paths are proper representatives of the ensemble of transition paths that would be obtained in an infinitely long simulation." seems misleading, given that it implies that the generate TPs are typical or highly likely to be observed.

In response, we changed this sentence to: “Here we resolve the Na⁺ and H⁺ transport cycle of PaNhaP by transition path sampling. The resulting molecular dynamics trajectories of ion exchange proceed without bias force, and overcome the enormous time-scale gap between seconds-scale ion exchange and microseconds simulations by importance sampling in path space.”

Please discuss the potential short-comings of the TP ensemble as generated here and preferably show evidence of how far generated paths are away from equilibrium.

For a discussion of the possible shortcomings and the evidence for the extent of sampling, please see the added or modified text in the two preceding responses.

4) Connection between simulations and experiments

The functional experiments for the I163A/Y255A double mutant are somewhat disconnected from the simulations and are used in a hand-waving manner to ascribe a function to the outer hydrophobic gate, namely to ensure fidelity at the expense of efficiency. I suggest the authors should test the same mutant in simulations and compare to experiment.

In response, we performed simulations of the I163A/Y255A mutant. The results showed that the mutated outside gate, while weakened, still prevents water access (Supplementary Fig. 13A). This finding, together with the earlier results, suggests that the I163A/Y255A remains functional, and requires less activation for the transition from an inward-facing to an outward-facing access state. We now write in Results: “To test this hypothesis we carried out

simulations with the I163A/Y255A double mutant in the inward-open conformation. The mutant shows conformational dynamics similar to the wildtype protein, albeit with a weakened outside gate (Supplementary Fig. 13A), consistent with the experimental data.”

MINOR COMMENTS

1) Modeling of an alternative conformation of a transporter by a procedure similar to the one used here was done (to my knowledge) for the first time by (using TMD towards another transporter)

S. A. Shaikh and E. Tajkhorshid. Modeling and dynamics of the inward-facing state of a Na⁺/Cl⁻ dependent neurotransmitter transporter homologue. PLoS Comput Biol, 6(8):e1000905, 2010.

and more recently (using MDFF directly) by

N. Coudray, S. Seyler, R. Lasala, Z. Zhang, K. M. Clark, M. E. Dumont, A. Rohou, O. Beckstein, and D. L. Stokes. Structure of the SLC4 transporter Bor1p in an inward-facing conformation. Protein Sci, 26(1):130–145, 2017.

I would consider citing Shaikh et al; the second reference is just to say that this computational protocol seems to become common.

We thank the Reviewer for alerting us of these references. We now write in Results: “Similar approaches of using homologous structures as reference for TMD have been applied to other transporters”, citing these references.

2) Please comment on the need to use MDFF with the cryo-electron crystallography density for the OF MjNhaP1 model. Why did the authors not simply use the published model with PDB ID 4D0A? How does their OF model differ from 4D0A? (Given the importance of the target conformation in the modeling procedure one would like to know what the features in the OF MjNhaP1 density are that ultimately determine the conformation of the PaNhaP structure.)

Early in the project, we indeed performed MD simulations using 4D0A as a target for TMD simulations. However, we found that the resulting structures were not stably committed to the outward-open state, returning back to the inward-open conformation in subsequent free equilibration run. We therefore decided to re-model the outward-open state based on the EM

map. We now write in Results: “However, using this structure (PDB ID: 4D0A) as reference for targeted MD (TMD) simulations to create an outward-open PaNhaP led to an incomplete conformational change.”

3) Some of the transition paths show that the ion binding site is sometimes connected to both compartments. Is this an indication of a leak for protons or Na⁺? --- water pathways have been observed for other transporters, see e.g.

J. Li, S. A. Shaikh, G. Enkavi, P.-C. Wen, Z. Huang, and E. Tajkhorshid. Transient formation of water-conducting states in membrane transporters. Proc Natl Acad Sci U S A, 110(19):7696–7701, Apr 2013.

so that wouldn't worry me. But what is the evidence that these water-wires could not transport protons?

We thank the Reviewer for alerting us of this reference. We now write in Results: “Note that transiently connected water wires between both sides of the membrane have been observed before in MD simulations of other transporters”, citing this reference. Two factors prevent ion and proton leakage. First, we noticed that with ions or protons bound in the binding site, we never observed another ion coming in the site. We attribute this to electrostatic repulsion. Second, with regards to proton transfer, the low concentration of protons (10^{-7} mol/l at pH 7) results in a very low frequency of proton encounters, on a 0.1 ms scale. With water spanning states forming very rarely and living at most for nanoseconds, the net flux of protons would be negligible. Together, the two effects effectively prevent ion and proton leakage. To clarify this issue, we added in Results: “A possible concern is that a water spanning pore could result in ion or proton leakage, which would short-circuit the ion exchange. However, with the ion binding site occupied we did not observe other ions entering. Moreover, the low proton concentration at pH 7 makes it exceedingly unlikely that a proton “arrives” during a short-lived pore-like intermediate state.”

4) Experiments by some of the authors [Wöhlert et al eLife (2014)] show that PaNhaP transport is cooperative at pH6 but not at pH5. This was taken as a major point in order to propose PaNhaP as a good model for human NHEs.

The experiments in this paper were performed at pH6. As mentioned above, it is not clear at which nominal pH the simulations are supposed to occur but assuming that protonation states

were chosen to be compatible with the experiments, cooperativity would be expected in the transport simulations. Namely, it should make a difference if both protomers are moving in parallel or if only one moves or if they moved in an anti-parallel fashion.

Please comment and at least make clear to the reader in how far the simulations address the remarkable experimental finding of cooperative transport.

We thank the Reviewer for raising this point. As now stated, our protonation states correspond to pH 7. In response to the question raised, we added in Discussion: “In our simulations, we could not extract information on the structural origin of the observed pH-sensitive cooperativity in ion transport, which remains unexplained.”

5) Can the authors use any of their data to approximate an underlying free energy landscape and a reaction-coordinate dependent diffusion coefficient? (Or is this impossible because the weights of the TP in TP ensemble are not known?)

Again, we thank the Reviewer for raising an excellent point. We indeed attempted to extract the free energy profile from the trajectories using a diffusion model. However, we could only construct very rough 1D models and do not feel comfortable to include the results in the paper.

Other point of the revision

Additional transition-path shooting simulations (20 ns x 50 trajectories) were conducted in the 3rd round following the second initial path to get more transition paths in this round. Updated figures and tables related this includes, Fig. 4C, Supplementary Fig. 6, Supplementary Table 1 and 2.

Reviewers' comments:

Reviewer #1 (Remarks to the Author):

I think the authors have put a significant effort to address some of my previous concerns. I think the study is important, but some of the answers to my previous queries warrant further clarification as to whether the paper describes relevant dynamics.

"In effect, path sampling allows us to "watch" the transport cycle unfold, with H⁺ and Na⁺ as substrates, both at near-ambient temperature (37°C) and at 100°C, the physiological temperature for *P. abyssi*"

This statement implies that the MD simulations capture the transport of ions across the membrane. However, in the reply to my question regarding ion-coupling the authors wrote that ion transport was a "rare" event and most of the time PaNhaP is in a resting state, facing either inwards or outwards (line 308). So, what is now unclear to me is whether the authors are actually monitoring ion-bound conformational changes by their MD simulations, i.e do they directly monitor ion transport by MD simulations? Please clarify because this is a critical point. If not, how can one assess if the conformational changes (structural trajectories) described by these targeted MD simulations are physiologically relevant?

The authors have not yet experimentally verified the hydrophobic inside gate (Ile69 and Ala132) by mutagenesis experiments. They have written however that the inner gate is less important because the residues are further apart. Maybe this is the case, but in the main paper (line 191) the authors write "Thus, inside and outside gates are an integral part of their ion-transport mechanism". These statements are contradictory. Also there is no distinction in the difference to the importance of the hydrophobic gates in the schematic Fig. 5. Since, the hydrophobic gates described in this paper are one of the new mechanistic findings, I think its additionally important that experiments are made to validate if these hydrophobic gates are really an integral part of the ion-transport mechanism.

Minor points:

Elevator and rocker-switch mechanisms describe the global conformational rearrangements of the transport domains. Local, gating events, such as the hydrophobic gates, are not described by these terms. As such, the following statement made by the authors does not make any sense: "Ion transport in PaNhaP thus integrates two seemingly exclusive models of transport by alternating

access: the elevator and the rocker switch". i.e, elevator proteins have gates too and their inclusion does not mean they work by a different alternating access model.

Reviewer #2 (Remarks to the Author):

I regret that the authors have not paid any attention to my comments. As such, it is very difficult for me to recommend publication.

In particular, I still disagree with the authors on the issue of similarity of the protein with NHE1. Their referral to earlier works by some of the same authors is unconvincing. The sequence alignments in those previous publications show that PaNhaP is as similar to the electrogenic NHaA as it is to electro-neutral NHE1. While they may share a handful of similar residues in the active site, the overall sequence similarity is utterly unnoticeable.

Similarly, protonation of Asp159 is key to the entire transport cycle. Ignoring it by stating that it is too costly to take into consideration is unacceptable. For example, the authors may choose to analyze the protein with a combined QM/MM simulation, alternatively, they may choose to examine different protonation states of the protein, or finally utilize empirical valence-bond (MS-EVB) methodology.

Finally, I appreciate that transition path sampling requires reaching the final state. Hence, I wish the authors would provide another means to validate their results other than simple repetition.

Reviewer #3 (Remarks to the Author):

The authors addressed all my concerns.

Response to Reviewers' comments

Reviewers' comments are shown in *Italic*.

Reviewer #1

I think the authors have put a significant effort to address some of my previous concerns. I think the study is important, but some of the answers to my previous queries warrant further clarification as to whether the paper describes relevant dynamics.

We thank the Reviewer for the encouraging comment.

*"In effect, path sampling allows us to "watch" the transport cycle unfold, with H⁺ and Na⁺ as substrates, both at near-ambient temperature (37°C) and at 100°C, the physiological temperature for *P. abyssi*"*

This statement implies that the MD simulations capture the transport of ions across the membrane. However, in the reply to my question regarding ion-coupling the authors wrote that ion transport was a "rare" event and most of the time PaNhaP is in a resting state, facing either inwards or outwards (line 308). So, what is now unclear to me is whether the authors are actually monitoring ion-bound conformational changes by their MD simulations, i.e do they directly monitor ion transport by MD simulations? Please clarify because this is a critical point. If not, how can one assess if the conformational changes (structural trajectories) described by these targeted MD simulations are physiologically relevant?

In response, we now make it clear that in our MD simulations, we monitored repeated ion transport events. We write in the abstract: "The resulting molecular dynamics trajectories of repeated ion transport events proceed without bias force." One of these ion transport events is shown in the Supplementary Movie 1. What might have been unclear in the previous response was our distinction between a spontaneous ion-transport event and an unbiased ion-transport trajectory. The former is unlikely to be simulated because it occurs spontaneously only on a seconds timescale whereas our simulations cover nanosecond to microsecond times. Now we add in the Discussion: "Thus, a spontaneous ion transport event is unlikely to be simulated within the computationally accessible timescale. This limitation was overcome by selectively simulating the

rare event with transition path sampling in this study.”

The authors have not yet experimentally verified the hydrophobic inside gate (Ile69 and Ala132) by mutagenesis experiments. They have written however that the inner gate is less important because the residues are further apart. Maybe this is the case, but in the main paper (line 191) the authors write “Thus, inside and outside gates are an integral part of their ion-transport mechanism”. These statements are contradictory. Also there is no distinction in the difference to the importance of the hydrophobic gates in the schematic Fig. 5. Since, the hydrophobic gates described in this paper are one of the new mechanistic findings, I think its additionally important that experiments are made to validate if these hydrophobic gates are really an integral part of the ion-transport mechanism.

We agree that we have strong evidence, from experiment and simulation, only for the “outside” hydrophobic gate between Ile163 and Tyr255. Therefore, we have removed any reference to a possible “inside” gate from the text, the schematic in Fig. 5 summarizing the mechanism, and SI. We now write in Results: “On the intracellular side, no tightly interacting hydrophobic gate residues were found. Ile69 and Ala132 interact weakly to control solvent access to the ion-binding site (Supplementary Fig. 11).” We correct the sentence the Reviewer pointed out and now write: “Thus, the hydrophobic gate between I163 and Y255 appears to be a conserved element of the ion-transport mechanism.” Consistent with the weak interaction between Ile69 and Ala132, we then show in the section entitled “Reaction coordinate for ion exchange from transition-path shooting” that the Ile69-Ala132 motion is not a significant element of the transport mechanism and write: “For the angle of the domain motion and the distance of the intracellular residue pair Ile69-Ala132 opening and closing, the coefficients take values around zero, which suggests no significant contribution from these order parameters to the transport mechanism.”

Minor points:

Elevator and rocker-switch mechanisms describe the global conformational rearrangements of the transport domains. Local, gating events, such as the hydrophobic gates, are not described by these terms. As such, the following statement made by the authors does not make any sense: “Ion transport in PaNhaP thus integrates two seemingly exclusive models of transport by alternating access: the elevator and the rocker switch”. i.e, elevator proteins have gates too and their

inclusion does not mean they work by a different alternating access model.

We thank the Reviewer for alerting us of this issue. The sentence pointed out by the Reviewer was removed. Now we write: “The dominant elevator-like 3-4 Å vertical movement of the transporter domain is thus associated with gate opening and closing.”

Reviewer #2

I regret that the authors have not paid any attention to my comments. As such, it is very difficult for me to recommend publication.

In particular, I still disagree with the authors on the issue of similarity of the protein with NHE1. Their referral to earlier works by some of the same authors is unconvincing. The sequence alignments in those previous publications show that PaNhaP is as similar to the electrogenic NHaA as it is to electro-neutral NHE1. While they may share a handful of similar residues in the active site, the overall sequence similarity is utterly unnoticeable.

We apologize that the previous references are not clear enough on this matter. Thus, we added a reference to a recent paper by Ben-Tal and colleagues which performed a very broad phylogenetic analysis of cation/proton antiporters [Masrati et al. Nat. Commun. 2018]. In this paper, they identified a motif that determines electrogenicity and ion selectivity. The electroneutral PaNhaP and NHE1 indeed share the same motif, which is different from the electrogenic NHaA. Although PaNhaP and NHE1 have the shared motif, the overall sequence can vary as the Reviewer implied. Thus, we restrict ourselves to mentioning the shared motif. Now we write in the Introduction: “Ben-Tal and colleagues recently reported a broad phylogenetic analysis of cation/proton antiporter families CPA1 and CPA2, and identified a motif that determines electrogenicity and ion selectivity... The shared motif determining electrogenicity and ion selectivity supports the use of PaNhaP and MjNhaP1 as model systems in mechanistic studies of electroneutral Na⁺/H⁺ exchange.”

Similarly, protonation of Asp159 is key to the entire transport cycle. Ignoring it by starting that it is too costly to take into considering is unacceptable. For example, the authors may chose to

analyze the protein with a combined QM/MM simulation, alternatively, they may choose to examine different protonation states of the protein, or finally utilize empirical valence-bond (MS-EVB) methodology.

We now clarify: “The only part of the cycle not accounted for is the quantum-mechanical simulation of (de)protonation of the presumed proton carrier Asp159 in the fully inward-open and outward-open states. However, the timescale of such simulations is at present orders of magnitude shorter than the MD simulations used in this study. Since our main aim is to study the ion shuttling mediated by large-scale protein conformational changes, thereby overcoming the enormous time-scale gap between seconds-scale ion exchange and microseconds MD simulations, the quantum-mechanical simulation of Asp159 (de)protonation is outside the scope of the present study. The (de)protonation of Asp159 was instead realized by switching between the deprotonated and protonated Asp159 in our MD simulations.” We note here that quantum mechanical simulations involving (de)protonation of protein residues at a reasonable level of theory (say, DFT/B3LYP) are currently limited to the pico- to nanoseconds regime. Multistate-EVB simulations have so far focused mostly on proton transfer mediated by water. In our case, MS-EVB would require extensive and highly challenging parametrizations for protein-mediated transfer in the conformationally variable environments presented by the inside and outside access states, and the buried binding site during the transition.

We do, however, realize that (de)protonation of Asp159 is an important part of the cycle, and performed MD simulations with deprotonated and protonated Asp159 in separate simulations (Supplementary Fig. 17). From these simulations, we showed that protonation and deprotonation of Asp159 are followed by unbinding and binding of sodium ion, respectively (Supplementary Fig. 17), confirming that Asp159 is responsible for both proton and sodium ion bindings. We now write in Discussions: “The observed ion-binding and structural characteristics in simulations of different protonation states of the residues in the ion-binding site are consistent with Asp159 being responsible for both proton and sodium ion bindings (Supplementary Fig. 17).”

Finally, I appreciate that transition path sampling requires reaching the final state. Hence, I wish the authors would provide another means to validate their results other than simple repetition.

As stated in the previous revision, we performed two independent transition-path-sampling

simulations starting from very different initial paths, and found robust features of paths such as domain motion, flipping of Asp159, and hydration state of the ion-binding site in the intermediate states. To clarify this point, we added a new figure showing a coupling between the domain motion and water-access state of the ion binding site from the two independent transition-path-sampling simulations (Supplementary Fig. 19). Now we write in Methods: “Robust features of transition paths were found from the two independent path-sampling simulations initiated from very different initial paths generated by TMD in opposite directions. In the transition paths, the domain motion changes the water-access state of the ion-binding site associated with a flipping motion of Asp159 (Supplementary Movie 1). The outward-open access state tends to emerge at a later stage of the domain motion toward the outward-open state (Supplementary Fig. 19). Considering that the two initial paths have different water-access-state profiles (Supplementary Fig. 19), the convergent behaviour demonstrates that the current method can find reliable and useful paths.”

Reviewer #3

The authors addressed all my concerns. — Oliver Beckstein

We thank the Reviewer for constructive reviewing.